# Unifi3D: A Study on 3D Representations for Generation and Reconstruction in a Common Framework

**Nina Wiedemann***     *nina.wiedemann@intel.com*
*Intel Corporation*

**Sainan Liu***     *sainan.liu@intel.com*
*Intel Corporation*

**Quentin Leboutet***     *quentin.leboutet@intel.com*
*Intel Corporation*

**Katelyn Gao**     *katelyng3@gmail.com*
*Intel Corporation*

**Benjamin Ummenhofer**     *benjamin.ummenhofer@intel.com*
*Intel Corporation*

**Michael Paulitsch**     *michael.paulitsch@intel.com*
*Intel Corporation*

**Kai Yuan**     *kai.yuan@intel.com*
*Intel Corporation*

**Reviewed on OpenReview:** *https://openreview.net/forum?id=GQpTWpXILA*

## Abstract

Following rapid advancements in text and image generation, research has increasingly shifted towards 3D generation. Unlike the well-established pixel-based representation in images, 3D representations remain diverse and fragmented, encompassing a wide variety of approaches such as voxel grids, neural radiance fields, signed distance functions, point clouds, or octrees, each offering distinct advantages and limitations. In this work, we present a unified evaluation framework designed to assess the performance of 3D representations in reconstruction and generation. We compare these representations based on multiple criteria: quality, computational efficiency, and generalization performance. Beyond standard model benchmarking, our experiments aim to derive best practices over all steps involved in the 3D generation pipeline, including preprocessing, mesh reconstruction, compression with autoencoders, and generation. Our findings highlight that reconstruction errors significantly impact overall performance, underscoring the need to evaluate generation and reconstruction *jointly*. We provide insights that can inform the selection of suitable 3D models for various applications, facilitating the development of more robust and application-specific solutions in 3D generation. The code for our framework is available at `https://github.com/isl-org/unifi3d`.

## 1 Introduction

Recent advancements in generative image synthesis architectures, such as Generative Adversarial Networks (GANs) and Diffusion Models, have driven significant progress in the field of 3D generation (Gezawa et al., 2020; Li et al., 2024; Zhao and Larsen, 2024; Liu et al., 2024a; Jiang, 2024). While image generation has

---

*Equal contribution

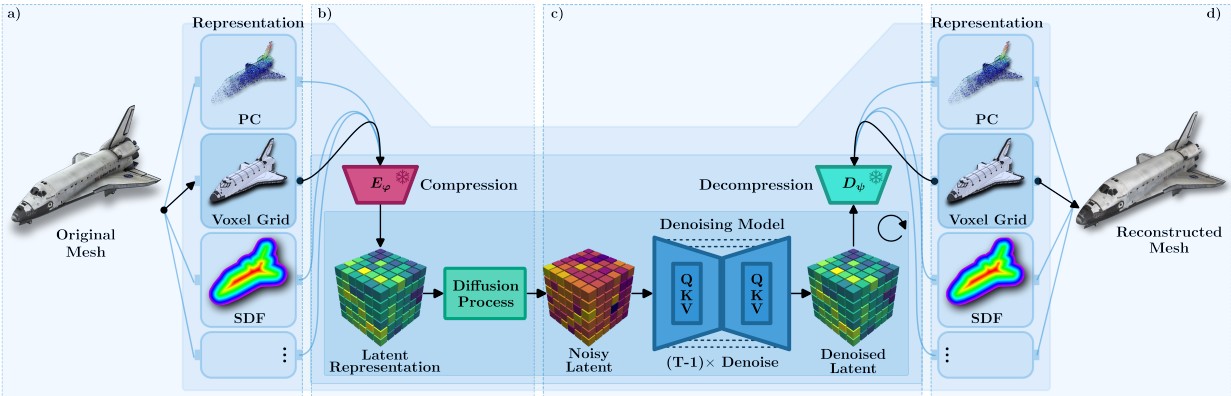

Figure 1: Overview of the steps involved in a standard 3D generation pipeline: a) the mesh is transformed into a suitable representation, b) an encoder $E_\varphi$ is pre-trained to compress it into a latent vector, c) a diffusion model is then trained to denoise the latent, d) the latent is finally decompressed into a target representation using a pre-trained decoder $D_\psi$ and turned back into a mesh using a dedicated algorithm such as marching-cubes. (PC: point cloud, SDF: signed distance field)

reached a stage of maturity, mainly standardizing around pixel-based representations (Crowson et al., 2024), the landscape for 3D representations remains fragmented and varied. A wide range of 3D representations, including Voxel Grids (Ren et al., 2024), Neural Radiance Fields (NeRFs) (Mildenhall et al., 2021), Signed Distance functions (SDFs) (Park et al., 2019), Point Clouds (PC) (Nichol et al., 2022), and Octrees (Wang et al., 2022b), have been proposed, each suited to different applications and tasks. These methods vary not only in the way they encode geometry but also in how they handle the trade-offs between quality, computational efficiency, memory requirements, and the ability to generalize to novel objects and scenes (Liu et al., 2024a; Peng et al., 2020; Wang et al., 2025).

While recent advances in 3D generative models can be attributed to various factors — such as the availability of improved datasets (Deitke et al., 2023), optimized sampling techniques for diffusion models (Ren et al., 2024), and enhanced generative architectures (Zhang et al., 2024) — the choice of the underlying representation remains a key factor. It dictates the information loss prior to encoding, influences the models used for compression and generation, and defines the computational resources for reconstructing a mesh from a generated sample. Therefore, assessing the suitability of different 3D representations for reconstruction and generation is of paramount importance.

However, objectively comparing 3D representations based on existing literature poses a significant challenge. While the representation is a core component, it is deeply embedded within complex 3D generative pipelines that utilize various models, loss functions, and datasets. On top of that, a wide variety of pre- and post-processing techniques is applied, significantly affecting the results. For instance, handling non-watertight meshes is often not well-documented, which tends to create inconsistencies across studies (Zhang et al., 2024). Another key issue lies in the difficulty of evaluating 3D object quality. Traditional metrics such as Chamfer Distance (CD) are commonly used to assess geometric accuracy, but they fall short in capturing perceptual quality and finer details (Mescheder et al., 2019; Wu et al., 2020). In turn, user studies can provide valuable qualitative insights, but they are labor-intensive and time-consuming. As a result, many papers resort to presenting qualitative results or cherry-picking favorable examples, which undermines objective evaluation and hinders progress in the field (Zhao and Larsen, 2024).

This paper introduces a unified evaluation protocol designed to benchmark 3D representations. We have developed a standardized pipeline that integrates all components of the generative process — from data preprocessing to mesh encoding, generation and mesh reconstruction — into a common framework, as illustrated in Figure 1. This design allows for the interchangeable use of 3D representations, ensuring that any observed performance differences are inherent to the representations themselves. Our standardized pipeline ensures a fair comparison between different representations with respect to reconstruction and generation,

instead of relying on the individual implementations of different 3D generation approaches that leverage the representations in varying manner, evaluate them with varying metrics, and oftentimes solely report conditional generation performance of the complete pipeline.

In contrast to *model* benchmarking, we aim to generate insights into the strengths and weaknesses of each representation while controlling for confounding factors like the diffusion model. Unlike traditional review papers that primarily offer qualitative overviews of existing methods (Po et al., 2024; Wang et al., 2025; Cao et al., 2024; Li et al., 2024; Gezawa et al., 2020; Liu et al., 2024a), our work emphasizes quantitative assessments grounded in empirical evidence. We make our entire pipeline open source to ensure the repeatability of our results and their generalization to different representations or experimental conditions. This allows rapid prototyping of novel 3D generative methods or 3D representations while ensuring adherence to best practices, such as proper preprocessing and hyper-parameter optimization. Our codebase also introduces previously unavailable open-source components, including a novel Dual-Octrees-based generative approach, training code for Shap-E, and transformer-based occupancy network generation. Summarized, the contributions of this paper are:

- **Standardized generation pipeline:** We implement a generation pipeline with plug-and-play functionality to test 3D representations (voxel grids, SDFs, point clouds, octrees, triplanes, and NeRFs) in a latent diffusion setting as the most common approach for 3D generation.

- **Jointly evaluating reconstruction and generation:** We analyze the relation between reconstruction and generation capability. Reconstruction errors are as high as 20% of the generation error, proving their significance.

- **Best practices:** We provide insights for common problems and derive best practices, such as the effects of different data preprocessing methods and the choice of meaningful sample sizes for evaluation.

- **Open-Source Modular Codebase:** We provide an open-source codebase with a structured, modular architecture (see Figure 1) for rapid development and evaluation of new methods with various generation models and 3D representations.

## 2   Benchmarking 3D Representations and Generative Algorithms

We put forward a framework for comparing *tensorial representations* of 3D objects. To ensure a fair and meaningful comparison, the following assumptions and requirements are established:

- **Meshes as ground truth:** The target representation in this study are *meshes*, due to their fundamental role in 3D computer graphics. They facilitate rapid rendering and are highly efficient in terms of space. Prominent datasets, such as ShapeNet (Chang et al., 2015) and Objaverse (Deitke et al., 2023), also provide objects as meshes.

- **Modularity:** The pipeline should be *modular*, allowing for the integration of different representations while controlling other components.

- **Coverage of related work:** The framework is designed to align with state-of-the-art methods in the field, accommodating the most significant representations utilized in recent research.

Based on these features, we propose a unified pipeline that captures the essential elements of contemporary 3D generation methods. As depicted in Figure 1, our multi-stage generation pipeline involves the following stages:

1. *Representation Conversion*: Transforming a 3D mesh into a suitable representation such as a voxel grid, SDF, or point cloud. This step often employs algorithmic transformations without requiring training.

2. *Representation Compression*: Utilizing an autoencoder to compress the high-dimensional representation into a lower-dimensional latent space using architectures like Autoencoder (AE) (Kim et al., 2023), a Variational Autoencoder (VAE) (Luo and Hu, 2021; Ren et al., 2024), or a Vector Quantized Variational Autoencoder (VQ-VAE) (Cheng et al., 2023).

3. *Latent Generation*: Training a generative model, typically a diffusion model (Sohl-Dickstein et al., 2015; Ho et al., 2020; Lyu et al., 2021; Zhou et al., 2021), to produce latent vectors that the decoder can reconstruct. If the representation is tokenizable, autoregressive models (Yan et al., 2022; Mittal et al., 2022; Siddiqui et al., 2023) may be employed instead.

4. *Mesh Reconstruction*: Reconstructing the final 3D mesh from the decoded representation, often using algorithms like Marching Cubes (Lorensen and Cline, 1987).

To support the claim that this pipeline is representative for most of the influential *direct* 3D generation approaches published over the past years, we list related work and their instantiations of representation, compressor and generator in Table 1. For instance, SDFusion Cheng et al. (2023) compresses an SDF-grid with a VQ-VAE and denoises the latent with a U-Net-based diffusion model; XCube Ren et al. (2024) encodes a voxel grid with a VAE and uses a multi-resolution U-Net diffusion; and Point-E Nichol et al. (2022) compresses a point cloud and denoises it via a Diffusion Transformer (DiT). For an in-depth discussion of related work, we refer to Appendix A.

## 2.1 Diffusion models

A suitable tensorial representation of 3D objects is part of any deep learning generative approach, whether it employs GANs, diffusion or autoregressive generation. Given their widespread use in the field, we opt for diffusion in our experiments; however, the main findings and identified shortcomings of certain representations are generic. We implement two diffusion models: a Diffusion Transformer (DiT) (Peebles and Xie, 2023) and a U-Net diffusion model (Rombach et al., 2022; Cheng et al., 2023). DiTs are adept at handling high-dimensional data by utilizing self-attention mechanisms and can be applied to any representation with an appropriate tokenization scheme, making them suitable for representations like point clouds and meshes. Conversely, U-Net architectures excel with grid-based representations such as voxel grids and SDFs, where the data can be organized into 2D or 3D tensors. For a detailed technical introduction to diffusion processes and for implementation details, see Appendix B.

## 2.2 Selected representations and architectures

The experiments in this study are conducted using a carefully selected set of 3D representations, chosen for several key reasons. First, we prioritized representations that are widely used in the field, ensuring the relevance of our analysis and alignment with existing literature. Second, we aimed to include a diverse range of representations, such as point clouds used as intermediate representation in the training process, and implicit representations like density fields and SDFs, which we use as the output representation to facilitate conversion to meshes for all methods. Third, scalability to large datasets and compatibility with diffusion models was a critical consideration, as this ensures that each representation can be effectively integrated with modern generative techniques. The encoders and decoders for each representation are trained through a reconstruction task on the same dataset used for the generative task. To ensure normally distributed latents, which is a crucial prerequisite for the diffusion process, the AE encoder always implements LayerNorm as its final layer. We implemented our representations, encoders, and diffusion models within a standardized training pipeline utilizing the `hydra` and `accelerate` libraries.

**Voxel and SDF Grid Encoding**  Voxel grids offer an intuitive and explicit encoding for 3D objects. While used predominantly in early research (Wu et al., 2015; Maturana and Scherer, 2015; Choy et al., 2016; Wu et al., 2016; Brock et al., 2016; Dai et al., 2018), they are still used in SOTA work (Ren et al., 2024; Zheng et al., 2023; Liu et al., 2023a). Instead of binary occupancy indicators, some works fill the grid cells with sampled SDF values (Cheng et al., 2023; Mittal et al., 2022) to increase the expressiveness. We implement both approaches: a standard voxel grid where each cell holds a binary occupancy value and an SDF grid where each cell contains the signed distance to the nearest point on the mesh surface, sampled

Table 1: Overview of representations, autoencoders and generators used in influential contemporary research on 3D generation (PC: Point cloud). Optimization-based methods are not included since they follow a different structure. Some methods do not compress the representation and directly denoise on the representation.

| Paper | Representation | Compression | Generation |
|---|---|---|---|
| gDNA (Chen et al., 2022) | Occupancy Field | - | GAN |
| BlockGAN (Nguyen-Phuoc et al., 2020) | Voxel Grid | - | GAN |
| XCube (Ren et al., 2024) | Voxel Grid | VAE | Sparse UNet diff. |
| Neuralfield-LDM (Kim et al., 2023) | Voxel | AE (CNN) | Hierarchical diff. |
| Trellis (Xiang et al., 2024) | Voxel / SparseFlex | Transformer VAE | Rectified Flow |
| LAS-Diffusion (Zheng et al., 2023) | Voxel / SDF | - | Diffusion UNet |
| One-2-3-45++ (Liu et al., 2023a) | Voxel / SDF | - | Diffusion UNet |
| Diffusion-SDF (Li et al., 2023) | Voxelized SDF | Patch-VAE | Diffusion UNet |
| LDM (Xie et al., 2024) | Voxelized SDF | - | Transformer |
| SDFusion (Cheng et al., 2023) | Voxelized SDF | VQ-VAE | Diffusion UNet |
| DualOctreeGNN (Wang et al., 2022a) | Octree / SDF | AE | |
| Make-A-Shape (Hui et al., 2024) | SDF | Wavelet features | Diffusion ViT |
| Cannonical mapping (Cheng et al., 2022) | PC | VAE | Autoregressive |
| DPM (Luo and Hu, 2021) | PC | VAE | Diffusion |
| PVD (Zhou et al., 2021) | Voxel / PC | - | Diffusion CNN |
| r-GAN / l-GAN (Achlioptas et al., 2018) | PC | AE | GAN |
| SoftFlow (Kim et al., 2020) | PC | AE | Normalizing Flow |
| DPF-Net (Klokov et al., 2020) | PC | AE | Normalizing Flow |
| Shape-GF (Cai et al., 2020) | PC / Density Field | AE | GAN |
| PointFlow (Yang et al., 2019) | PC | VAE | Normalizing Flow |
| PointGrow (Sun et al., 2020) | PC | MLP | Autoregressive |
| 3dAAE (Zamorski et al., 2020) | PC | VAE | AAE |
| tree-GAN (Shu et al., 2019) | PC | - | GAN |
| Point-E (Nichol et al., 2022) | PC | Coordinates | DiT |
| 3DShape2VecSet (Zhang et al., 2023) | PC / SDF | Transformer | DiT |
| CLAY (Zhang et al., 2024) | PC / SDF | Transformer | DiT |
| Michelangelo (Zhao et al., 2023) | PC / Occupancy | VAE | Diffusion UNet |
| SparseFlex (He et al., 2025) | PC / SparseFlex | Transformer VAE | Rectified Flow |
| AutoSDF (Mittal et al., 2022) | SDF | VQ-VAE | Autoregressive |
| SurfGen (Luo et al., 2021) | SDF | - | GAN |
| 3D-LDM (Nam et al., 2022) | SDF | MLP | Diffusion (MLP) |
| TripoSG (Li et al., 2025) | SDF | Transformer VAE | Rectified Flow |
| SDM-NET (Gao et al., 2019) | Mesh | - | VAE |
| PolyGen (Nash et al., 2020) | Mesh | Coordinates | Autoregressive |
| MeshGPT (Siddiqui et al., 2023) | Mesh (face tokens) | GraphAE | Autoregressive |
| MeshXL (Chen et al., 2024b) | Mesh | Coordinates | Autoregressive |
| GIRAFFE (Fu et al., 2022) | NeRF | - | GAN |
| HoloDiffusion (Karnewar et al., 2023) | NeRF | ResNet | Diffusion |
| Shape-E (Jun and Nichol, 2023) | NeRF / SDF | Transformer | DiT |
| SSDNeRF (Chen et al., 2023a) | NeRF / Triplane | MLP | DiT |
| EG3D (Chan et al., 2022) | Triplane | - | GAN |
| 3DGen (Gupta et al., 2023) | Triplane | PointNet/Unet | Diffusion UNet |
| Direct3D (Wu et al., 2024a) | PC / Triplane | - | DiT |
| Get3D (Gao et al., 2022) | Triplane+DMTet | - | StyleGAN |
| TriFlow (Wizadwongsa et al., 2024) | Triplane | MLP | DiT |
| ShapeFormer (Yan et al., 2022) | PC / VQDIF | Transformer | Autoregressive |

from a truncated SDF with a cutoff of 0.2. Meshes were converted to 3D grids of resolution $64^3$ by ray casting and closest point queries. For encoding, we implemented a 3D CNN following Cheng et al. (2023). For example, a grid of resolution $64^3$ is transformed into a latent tensor of shape $3 \times 16 \times 16 \times 16$. This latent representation is typically diffused using a 3D U-Net (Cheng et al., 2023). Here, we additionally introduce a transformer-based approach, where the latent tensor is tokenized by dividing it into 3D patches of size $4^3$, analogous to patch-based tokenization used in vision transformers. This results in a total of 12 potential

generative approaches: Voxel / SDF grid, each combined with AE / VAE / VQ-VAE, and denoised with DiT or U-Net.

**3DShape2VecSet Encoding**   Occupancy Networks (Mescheder et al., 2019) and their extensions (Peng et al., 2020; Atzmon and Lipman, 2020; Zhang et al., 2023) use implicit representations to sample density or occupancy at any spatial point, enabling detailed surface generation beyond grid limitations. This method has recently been leveraged for conditional 3D generation, achieving remarkable results (Zhang et al., 2024; Yang et al., 2024; Li et al., 2025). Our implementation, based on the 3DShape2VecSet model (Zhang et al., 2023), transforms meshes into point clouds $X$ and sub-samples them ($\hat{X}$). The positional embeddings of the points are processed through cross-attention between $X$ and $\hat{X}$, resulting in a set of $k$-dimensional latent vectors. The decoder passes the vector set through multiple self-attention layers, with cross-attention applied between the outputs and embedded query points. Following Zhang et al. (2023), latent generation employs a DiT.

**Dual Octree Graph Encoding**   To efficiently capture high-resolution geometric details while maintaining computational scalability, we implemented the dual octree graph representation proposed by Wang et al. (2022b). This method leverages the hierarchical structure of octrees combined with graph neural networks (GNNs) to effectively represent complex 3D geometries. The dual octree graph network takes a set of point clouds as input, builds a dual octree graph, and outputs an adaptive feature volume via a graph-CNN-based encoder-decoder network structure. In all experiments, we use a tree depth of 6 and force the octree to be full when depth $d \leq 2$. During decoding, the encoded Octree-features are converted to a signed distance field via a multilevel neural partition-of-unity (Neural MPU) module to construct a surface. Our approach extended this AE structure to VAE and VQ-VAE and trained a simple two-layer 3D U-Net diffusion model for generation. To the best of our knowledge, this is the first generative approach based on Dual Octrees.

**Triplanes**   Triplane representations efficiently encode 3D scenes in a *hybrid explicit-implicit* manner Wang et al. (2025), utilizing three orthogonal 2D feature planes Peng et al. (2020). From these planes, a lightweight MLP occupancy network can perform volumetric rendering. We employ the autoencoder architecture proposed in Peng et al. (2020) to generate triplane latents from point clouds sampled on the mesh. Following Wu et al. (2024b), latent generation employs a UNet with 3D-aware convolution layers to better capture the unique properties of the triplane latents.

**NeRFs**   Neural radiance field (NeRF) is another implicit representation that was first proposed for novel view synthesis Mildenhall et al. (2021), parameterizing the color and density at each point by a neural network. To render the 3D object from novel views, a differentiable volumetric ray casting approach is utilized. The vast majority of generative methods using a NeRF representation are optimization-based, iteratively updating the parameters of the neural network to maximize the likelihood of rendered images under an image generation model Poole et al. (2022); Lin et al. (2023); Liu et al. (2023b); Wang et al. (2024); Di Sario et al. (2024). Therefore, in this work, we adopt instead the NeRF representation used in Shap-E Jun and Nichol (2023), which falls into the framework illustrated in Figure 1. Each mesh is converted into a point cloud and multi-view RGB images, which are encoded into a latent vector using point convolution, cross attention, and a Transformer. To ensure that the latent vector is bounded, it is input into a tanh activation function, and noise is added; however, to be consistent with the other representations, we replace this step with layer normalization. During decoding, the latent vector is projected to the parameters of the NeRF's neural network. The autoencoder is trained on a combination of RGB and transmittance reconstruction losses on multi-view renderings. In the original paper, the latent vector is also projected to the parameters of a signed texture field, and the autoencoder is fine-tuned with SDF and color distillation losses. However, we exclude this step to have a fair comparison of NeRF with the other representations. As in the original paper, we leverage DiT for generation. For both reconstruction and generation, meshes are obtained from the NeRF by applying Marching Cubes to its density function.

### 2.3 Evaluation protocol: Joint evaluation of reconstruction and generation

The ability to transform a mesh to and from latent space upper-bounds a representation's generation capabilities, as generated vectors must ultimately be converted into meshes. Thus, we advocate for jointly benchmarking reconstruction and generation to understand how reconstruction errors limit generation. Reconstruction error comprises representation-inherent errors, like information loss in voxel grids and compression errors due to the encoder. Additionally, an encoder trained on limited data may struggle with novel objects, necessitating an assessment of out-of-distribution (OOD) performance. Consequently, we propose the following evaluation protocol:

1. Assess reconstruction (mesh → representation → mesh)
2. Benchmark compression performance (mesh → representation → latent vector → representation → mesh).
3. Test the encoder's generalization in OOD tasks
4. Measure generation performance quantitatively and qualitatively.

## 3 Experimental setup

We benchmarked the selected methods on a subset of categories of the ShapeNet dataset (Chang et al., 2015), specifically *car*, *airplane*, and *chair*, following Ren et al. (2024). The dataset was split into training, validation, and test sets according to the official division[1]. We evaluated reconstruction performance on the full test dataset (806 objects for *airplane*, 703 for *car*, and 1,163 for *chair*). Following Peng et al. (2020), we assessed reconstruction quality using Chamfer Distance (CD), F-Score, and Normal Consistency (NC). For precise definitions, please refer to Appendix C. The selected metrics can be directly computed on the meshes and, unlike metrics like Intersection over Union (IoU), do not assume meshes being manifold, watertight, or convertible to grid representations — assumptions that do not hold for many meshes in the dataset.

Quantitatively evaluating generative models for 3D shapes is challenging due to the diversity of tasks and metrics in related work, such as single-image or multi-view reconstruction, text-to-3D synthesis, or unconditional generation. To ensure a fair comparison and to avoid adding complexity to our benchmarking pipeline, we opted to evaluate the models in an unconditional generation setting. This approach tests the ability of the representations to generate objects that are both similar to the training set and diverse. Following previous work (Gao et al., 2022; Lei et al., 2023; Yang et al., 2019), we implemented three distributional metrics: Coverage (COV), Minimum Matching Distance (MMD), and 1-Nearest Neighbor Accuracy (1-NNA). All these metrics are based on the pairwise CD between a set of generated samples $S_g$ and a reference dataset $S_r$, which is a subset of the test data. Coverage measures the fraction of $S_r$ that is matched to $S_g$, reflecting the diversity of the generated samples (higher is better). MMD computes the average minimum distance from samples in $S_r$ to those in $S_g$, indicating the quality of the generated samples (lower is better). The 1-NNA metric assesses the overfitting of the model by measuring the accuracy of classifying samples into $S_r$ and $S_g$ based on their distances. An ideal model achieves a 1-NNA of 0.5, indicating that generated samples are indistinguishable from real data. For formal definitions of these metrics, see Appendix C. In Figure 2, we measure the stability of these metrics by computing them on random subsets of ShapeNet, imitating a perfect generative model. The meaningfulness of the metrics strongly depends on the chosen set size for $S_r$ and $S_g$, making it difficult to compare numbers in the literature. For instance, the MMD metric decreases with larger set sizes as the pool of meshes to find the closest neighbor increases. We also observe a large spread of values for small set sizes and recommend that the set size should be larger than 200 for the metrics to converge.

To evaluate the subjective mesh quality, we conduct a user study. In each question, two meshes generated by different approaches are presented to the user, who is asked to indicate which is preferable based on object complexity and surface quality. Overall, we collect a dataset of 575 preferences from 24 users. To obtain scores for each approach, we model the preferences using the Bradley-Terry statistical model (Bradley and Terry, 1952) (see Appendix J for details).

---

[1] `http://shapenet.cs.stanford.edu/shapenet/obj-zip/SHREC16/all.csv`

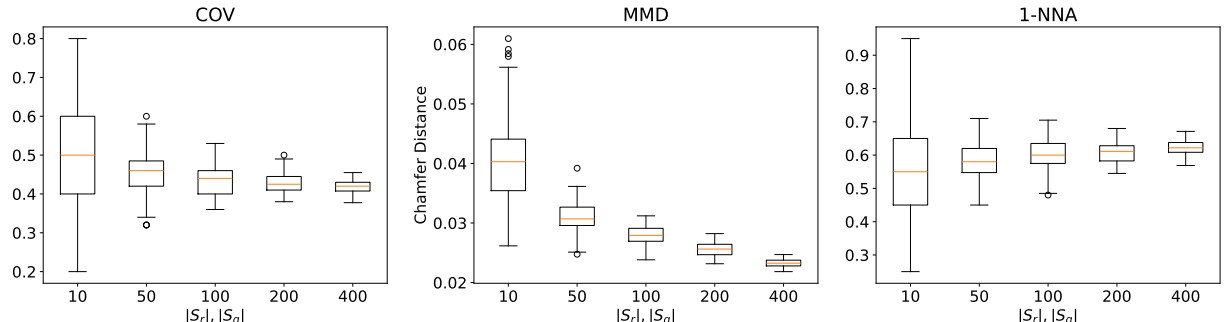

Figure 2: We select 100 random subsets of different sizes of the train and test split of the ShapeNet airplane category and compute the COV, MMD, and 1-NNA metrics. Small set sizes result in a large spread of values. We further observe that the Coverage (COV) is not close to the optimal value 1.0 and the 1-Nearest Neighbor Accuracy (1-NNA) is above the optimal value 0.5, indicating that learning from the training set may lead to biases.

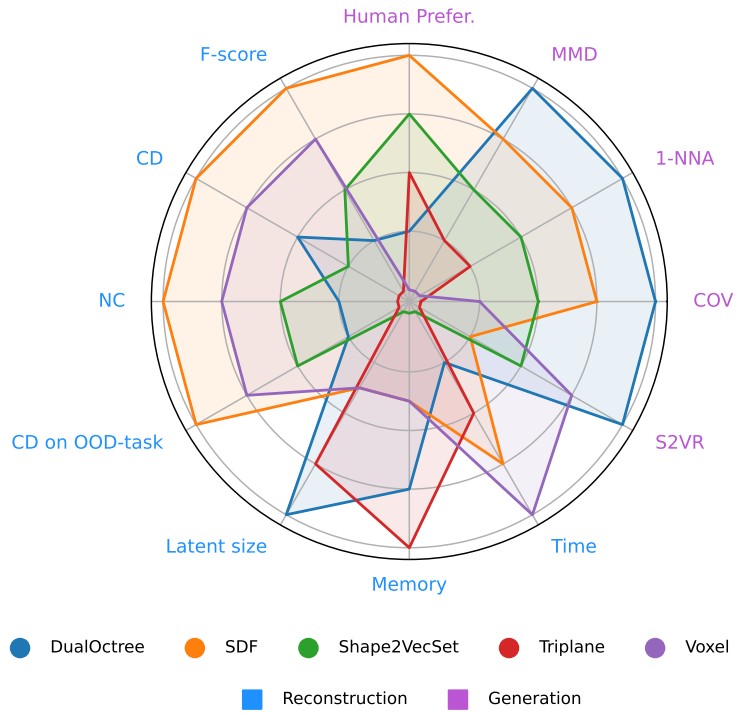

Figure 3: Rankings of 3D representations based on generation and reconstruction metrics. The outer circle indicates the top rank. This chart compares reconstruction metrics (CD, F-score, NC), reconstruction generalization on an OOD task, and reconstruction efficiency (memory footprint, encoding size, and inference time) as well as the metrics for unconditional generation performance with user study rankings (1-NNA, MMD, COV, surface-to-volume ratio (S2VR), Human Preference).

## 4 Results and Discussion

Figure 3 summarizes the results, demonstrating that the SDF-encoder achieves the best reconstruction performance, but our DualOctree diffusion model excels in generation. In the following, the results will be discussed step-by-step.

### 4.1 Benchmarking generation performance

The comparison of the best generative approaches, one per representation, is shown in Table 2. For ablation studies on encoders, i.e. selecting the best from AE / VAE / VQVAE, and on diffusion models (DiT / UNet), see Appendix G. Interestingly, we found approaches with AE encoders usually outperforming VAEs, contrary to prior findings. Apparently, adding layer normalization to AE has a similar positive effect as the

KL-loss in VAE. Surprisingly, the novel DualOctree-based diffusion model achieves best performance in all metrics, followed by SDF and Shape2VecSet. This finding contrasts with the widespread use of Shape2VecSet in state-of-the-art (SOTA) research, highlighting the critical role of multi-scale input schemes and the larger transformer models employed in recent SOTA developments Zhang et al. (2024).

Table 2: Performance of representations in unconditional generation setting.

| Method | COV ↑ | MMD ↓ | 1-NNA → 0.5 |
|---|---|---|---|
| DualOctree VAE UNet | **0.365** | **0.031** | **0.824** |
| SDF AE DiT | 0.357 | 0.032 | 0.860 |
| Shape2VecSet | 0.344 | 0.033 | 0.864 |
| Triplane AE UNet | 0.297 | 0.036 | 0.921 |
| Voxel AE DiT | 0.319 | 0.040 | 0.937 |

However, the user study (see Figure 3) paints a different picture, ranking SDF (score 0.25), Shape2vecset (-0.122) and Triplane (-0.140) above DualOctree (-0.178). This shows the necessity to run a user study when aiming to assess human-perceived object quality. One reason for the users' preference for SDF-generated objects may be the tendency of SDF-grids to generate smooth surfaces. Figure 4 provides qualitative results for the generative approaches. On the other hand, we found that DualOctree generates more *complex* – but potentially imperfect – assets, measured in terms of the surface-to-volume ratio. Appendix I expands on the analysis of complexity and provides further evidence that SDF tends to smoothen surfaces whereas Shape2VecSet and DualOctree are more prone to creating artifacts. Furthermore, Appendix E provides the corresponding results on a subset of the Objaverse dataset, demonstrating similar results but better performance of the Triplane representation.

## 4.2 Reconstruction performance

Table 3 provides the reconstruction quality in terms of CD, F-Score, and NC. The best-vectorized representations are the voxel and SDF grid encodings using AE and VAE, with an average F-score of 88%. NeRF-encoding performs worst, probably due to modifications in our implementation for the sake of comparability (see section 2.2), and NeRF-based generation did not reach comparable generation performance and we thus only analyze its reconstruction performance. Ablation studies with respect to the encoder model (AE / VAE / VQ-VAE) are provided in Appendix F.

Furthermore, we hypothesized that larger latent representations reduce the information loss during compression but come along with longer runtimes. Figure 5 shows the trade-off between reconstruction loss, reconstruction runtime, and the size of the latent vector (in terms of the number of elements of the tensor). The results shows well the different characteristics and importance of different representations. For instance, DualOctree with the smallest latent size achieves better reconstructions than Triplane with a larger latent thanks to the spatial adaptive structure but is less accurate and slower than the Voxel representation that uses a much larger latent. Selecting the right representation can therefore significantly influence the runtime, accuracy and memory requirements. Generative methods targeting CAD applications may want a representation with minimal reconstruction error, while methods running on devices with small memory may favor a representation with a small latent.

## 4.3 Generalization ability of the encoder

When encoding 3D meshes into low-dimensional vectors, there is a trade-off between the compression capacity and the model's capability to generalize to new representations. Since the generator aims to design new objects, applicability beyond the training data is crucial. We quantified the generalization capability (see Table 3 - right) where we test the encoders trained on the *Chair* category when applied to the *Airplane* category. While most encoders show very high generalization performance, on par with category-specific training, NeRFs and DualOctrees stand out as representations that struggle with OOD data. In the former, the MLP-based latent may be prone to overfitting, while for the latter, the low dimensionality of the latent could be problematic.

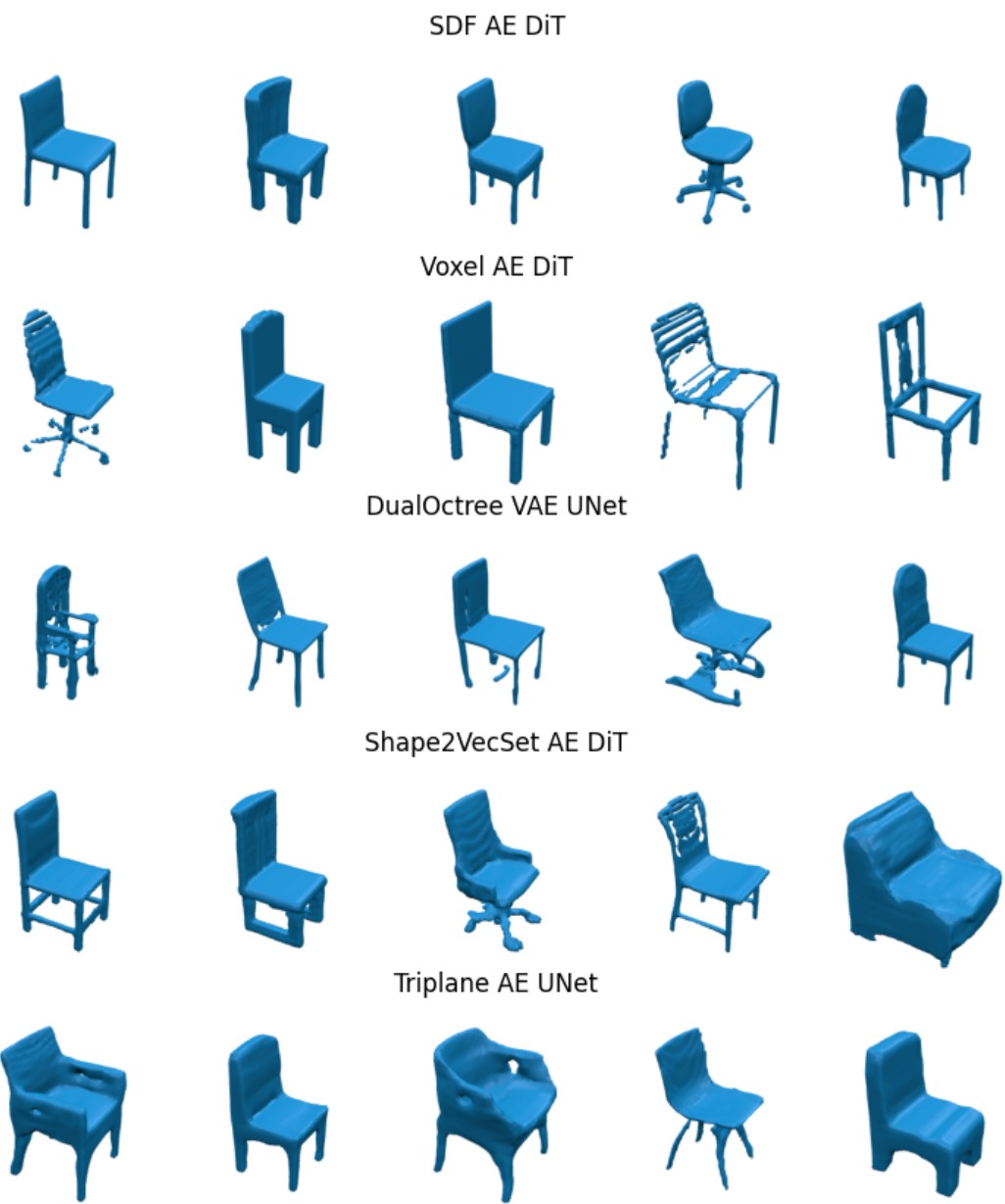

Figure 4: Qualitative results for mesh generation. We show results for each representation using the best encoder configuration for each.

Table 3: Reconstruction quality in terms of CD, F-score and NC.

| Method | Trained & tested on Airplane / Car / Chair | | | OOD (trained on Chair, tested on Airplane) | | |
|---|---|---|---|---|---|---|
| | F-score (0.0125) ↑ | CD (*1e-4) ↓ | NC ↑ | F-score (0.0125) ↑ | CD (*1e-4) ↓ | NC ↑ |
| DualOctree VAE | 76.122 ± 13.44 | 0.02 ± 0.01 | 0.766 ± 0.07 | 48.38 ± 11.39 | 0.047 ± 0.02 | 0.677 ± 0.08 |
| NeRF AE | 58.44 ± 13.22 | 0.034 ± 0.02 | 0.723 ± 0.07 | 26.229 ± 11.64 | 0.107 ± 0.04 | 0.589 ± 0.05 |
| SDF AE | **88.434 ± 6.58** | **0.012 ± 0.0** | **0.827 ± 0.06** | **91.123 ± 6.02** | **0.01 ± 0.01** | **0.843 ± 0.05** |
| Shape2VecSet AE | 79.37 ± 17.04 | 0.023 ± 0.02 | 0.776 ± 0.07 | 75.338 ± 8.87 | 0.022 ± 0.01 | 0.717 ± 0.07 |
| Triplane AE | 66.445 ± 16.06 | 0.028 ± 0.02 | 0.759 ± 0.08 | 41.69 ± 11.57 | 0.073 ± 0.03 | 0.688 ± 0.07 |
| Voxel AE | 85.666 ± 10.54 | 0.016 ± 0.01 | 0.787 ± 0.06 | 85.602 ± 9.48 | 0.017 ± 0.01 | 0.8 ± 0.05 |

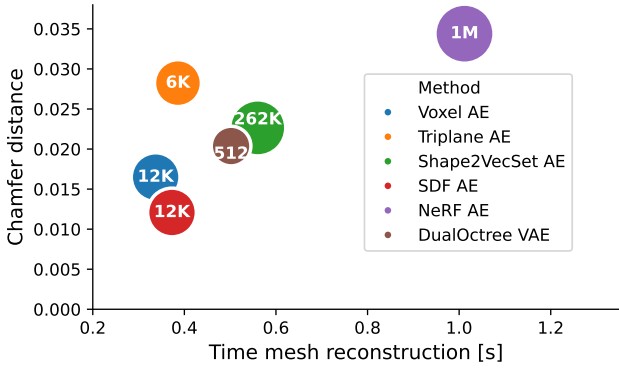

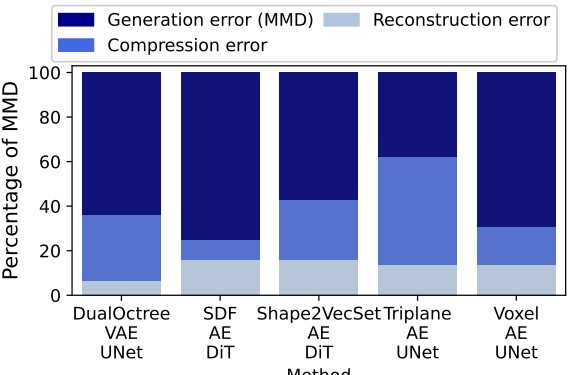

Figure 5: Reconstruction quality by runtime for inference (decoding and mesh reconstruction), and by size of the latent (see label and bubble size).

Figure 6: Decomposition of errors during generation. Reconstruction errors and imperfect decompression play a substantial role.

## 4.4 Effect of preprocessing

For all methods — except those encoders operating directly on point clouds or meshes — preprocessing meshes is essential. Non-watertight meshes introduce significant artifacts as they do not have a well defined inside or outside, which is crucial information for many methods, adversely affecting the quality of the reconstructed shapes. Mesh manifoldization is an active research area, with various methods proposed like the *Manifold* library (Huang et al., 2018), which thickens surfaces to ensure that every mesh component forms a solid volume. While effective, this process can alter the original geometry by artificially expanding thin structures. *ManifoldPlus* (Huang et al., 2020) extends *Manifold* but has been reported to produce inconsistent results (Zhang et al., 2024). Wang et al. (2022b) uses a simple manifoldization (Mesh2SDF) based on contouring the unsigned distance function, which also produces significant thickening artifacts or requires high grid resolutions, resulting in high compute and memory requirements. The thickening of the mesh helps to preserve thin structures but adds an irrevocable bias. To this end, we introduce a lean and mean preprocessing step that transforms meshes to SDFs without thickening. Figure 7a shows a visual explanation of our preprocessing step using the flood fill algorithm. Instead of altering the mesh we define the inside and outside as outlined in Figure 7a:

(1) We define a target grid encompassing the mesh.
(2) We mark all voxels that touch the mesh surface (gray) and then flood fill starting from a corner voxel guaranteed to be outside to define the outside region (blue).
(3) Voxels that touch the mesh but not an outside voxel defined by the 26-voxel-neighborhood get removed. This step effectively eliminates internal structures not considered part of the outer shape of an object.
(4) All unlabelled voxels get labelled as *inside* (red).
(5) For all gray voxels we determine whether the voxel center lies inside or outside by comparing to the plane defined by the closest point to the surface and the normal approximated by the sum of

the positions of the outside neighbor voxels. The surface points sampled in this step are reused for computing distances for the SDF.

Finally, we directly sample points and compute distances with the determined sign.

We compare the effects of manifoldization as a preprocessing step by converting meshes to a grid SDF representation and back to the mesh representation in Figure 7b (see Appendix H for additional results).

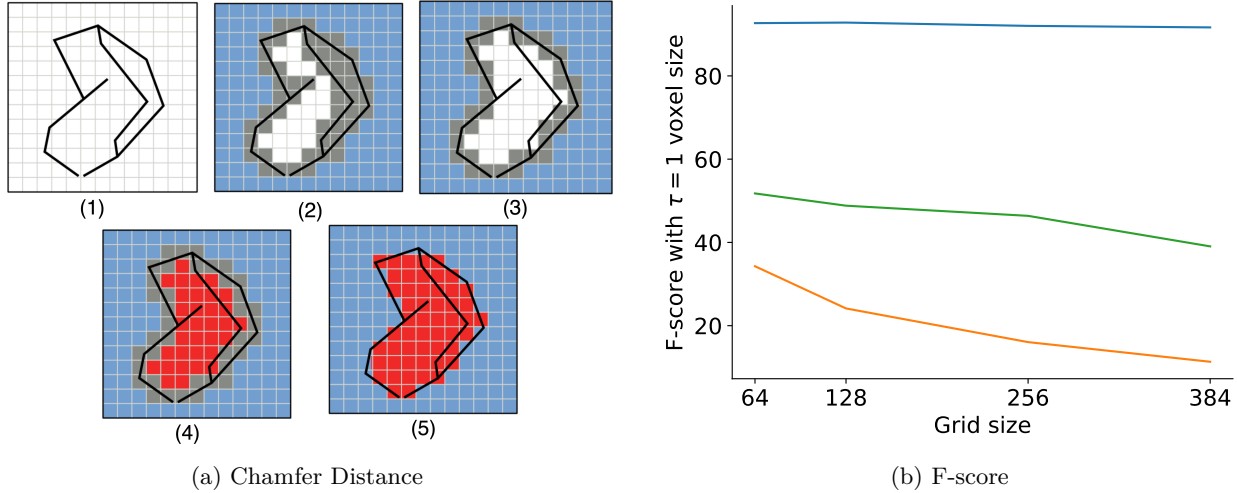

(a) Chamfer Distance                                    (b) F-score

Figure 7: We compare different methods for converting a mesh to an SDF. **Ours** uses the flood fill approach to define the inside (red) and outside (blue) of a mesh, as explained in a. **Naive** uses a simple raycasting approach to determine the sign. **Mesh2SDF** from Wang et al. (2022a) creates a watertight mesh using the unsigned distance and suffers from artificial thickening of the shape. This effect is reduced with increasing grid sizes.

### 4.5   Error decomposition

In alignment with the goal to evaluate reconstruction and generation *jointly*, we investigate the relation between generation, compression, and reconstruction errors. Since MMD measures generation performance as the mean CD between each sample in $S_r$ and its best match in $S_g$, it can be compared to the CD at the reconstruction or compression stage. While the generation of new shapes inevitably induces an MMD $> 0$ (see Figure 2), the MMD is loosely lower bounded by the reconstruction and compression error in terms of CD. To decompose the error, for each sample $\rho \in S_r$, we compare its MMD, its compression error (e.g. encoding-decoding), and its mesh reconstruction error (e.g. marching cubes inaccuracies). The reconstruction errors are taken from the analysis in Figure 7b while the compression errors correspond to the sample-wise result from Table 3. It is worth noting that these errors are not simply additive; however, it is interesting to investigate the size of the lower bounds (reconstruction and compression errors) in relation to the absolute size of the generative error. Figure 6 shows that the reconstruction and compression errors amount to 12.9% and 39.3% of the MMD, respectively, when averaging over all representations. The MMD and the reconstruction CD correlate on average with a Pearson R correlation of 0.30 ($p < 0.01$). As expected, there is also a positive correlation of $r = 0.24$ between compression and reconstruction error. This analysis underlines the important role of 3D representations and their reconstruction performance, as reconstruction errors can significantly reduce the quality of the generated samples.

## 5   Conclusion

We have presented a systematic comparison of 3D representations for reconstruction and generation. Our analysis leads us to recommend the following best practices:

- Evaluate reconstruction and generation jointly. Since generation quality is upper-bounded by the reconstruction error, the errors resulting from mesh reconstruction and representation encoding alone should be reported for transparency.

- Compute the errors with respect to the original mesh. Results should not be distorted by derivatives obtained following the application of methods like *Manifold*[2], and avoid metrics that require additional preprocessing.

- Include conversions of the output representation to meshes. Comparing only generated surface points to the ground truth mesh may skew results.

- A sufficient number of samples is crucial when evaluating unconditional generation with metrics such as MMD, 1-NNA, and Coverage. More than 200 samples are generally necessary to achieve robust outcomes.

**Limitations**   This framework covers *direct* 3D generation, in contrast to optimization-based methods such as *Score Distillation Sampling (SDS)* -based methods (Poole et al., 2022; Chen et al., 2023b; Lin et al., 2023; Sun et al., 2023; Wang et al., 2024; Shi et al., 2023b) or other approaches that use multi-view images for inference Shi et al. (2023a); Liu et al. (2024c); Long et al. (2024). Despite the impressive results generated with these techniques, recent developments have shifted the focus of the field back to direct 3D generation, due to 1) high computational costs of inference-time optimization Li et al. (2024), 2) dependence of the generation quality on the fidelity of the multi-view images Wu et al. (2024a), 3) preferability of explicit 3D outputs for artists Yang et al. (2024), and 4) feasibility of training for direct generation due to large-scale 3D datasets such as Objaverse Deitke et al. (2023) and ObjaverseXL Deitke et al. (2024). Similarly, Gaussian Splatting has become popular in the 3D field Tang et al. (2023); Kerbl et al. (2023); Lan et al. (2024), but is not included in our study, as it is usually trained with SDS-loss methods and focuses on realistic and efficient rendering, in contrast to our assumption of meshes as the ground truth.

Furthermore, some generative methods fit into our unified pipeline but remain unimplemented, such as autoregressive generation. Follow-up work could also extend the analysis to other datasets. While we already provide results on a subset of Objaverse in Appendix E, further experiments distinguishing different types of objects (e.g. thin structures or very complex shapes) could bring further insights on the advantages and drawbacks of each representation.

**Outlook**   There are several open challenges in the field, such as accounting for interior structure of objects, transferring 3D *object* generation methods to a *scene* level Ren et al. (2024), or enabling model articulation (Leboutet et al., 2024; Lei et al., 2023; Liu et al., 2024b). While our benchmark shows the general ability of any representation to encode and generate high-resolution objects, it also shows the challenges such as dealing with thin structures while retaining computational efficiency. With the presented pipeline implemented in an open-source code base, we offer a unified framework to develop and benchmark novel 3D representations for generation.

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

## Appendix

## A  Representations and Algorithms in 3D Generative Modeling

Meshes are foundational in 3D computer graphics and vision, representing surfaces in terms of vertices, edges, and faces to form polygonal approximations of objects. While meshes are extensively used across various fields due to their broad compatibility with rendering pipelines and ability to depict complex geometries with high fidelity, processing them within neural network architectures presents significant challenges. The variable topology and connectivity inherent in meshes make them difficult to process using standard neural architectures that expect regular input structures. Consequently, processing meshes often requires specialized neural networks, such as graph neural networks (Li et al., 2024), or autoregressive architectures (Yan et al., 2022; Mittal et al., 2022; Siddiqui et al., 2023) which can be complex and computationally inefficient. Moreover, high-resolution meshes can have substantial memory footprints, raising concerns about the scalability of the resulting learning pipelines. These limitations have led researchers to explore alternative three-dimensional representations derived from meshes through algorithmic conversions that typically do not require training to facilitate processing, learning, and rendering within deep neural frameworks Li et al. (2024).

**Voxel grids**  Voxel grids offer an intuitive and straightforward encoding for 3D objects analogous to pixel representations in 2D images. Early research predominantly relied on dense voxel grids (Wu et al., 2015; Maturana and Scherer, 2015; Choy et al., 2016; Wu et al., 2016; Brock et al., 2016; Dai et al., 2018). Despite their convenience, voxel grids suffer from a large memory footprint, limiting the achievable resolution and computational efficiency. Recent methods have revisited voxel grids to leverage their compatibility with convolutional neural networks (CNNs). For instance, X-Cube (Ren et al., 2024) introduces a multi-resolution approach involving denoising and decoding steps, while methods like LAS-Diffusion (Zheng et al., 2023) and

One-2-3-45++ (Liu et al., 2023a) utilize voxel grids in the initial stages of their generation pipelines. These approaches demonstrate that voxel grids can still be effective when combined with modern techniques like diffusion models. To mitigate memory constraints, sparse or hierarchical voxel grids have been proposed (Liu et al., 2020; Fridovich-Keil et al., 2022; Sun et al., 2022). Notably, Instant Neural Graphics Primitives (Instant-NGP) (Müller et al., 2022) use a multi-level voxel grid encoded via a hash function, enabling fast optimization and rendering while maintaining a compact model size.

**Implicit neural functions**   Implicit representations model 3D geometry as continuous functions, allowing for high-resolution detail without the memory overhead of dense grids. Occupancy Networks (Mescheder et al., 2019) introduced an implicit representation that predicts the occupancy probability of arbitrary points in 3D space, conditioned on input data such as images or point clouds. This approach allows for smooth and detailed surface representations. Extensions of this idea include Convolutional Occupancy Networks (Peng et al., 2020) and Neural Implicit Surfaces (Atzmon and Lipman, 2020), which improve the ability to capture fine geometric details. Recently, Zhang et al. (2023) proposed 3DShape2VecSet, employing a Transformer-based encoder and decoder where occupancy at query points is predicted via cross-attention mechanisms. This representation and follow-up work Chen et al. (2024a) have been used for conditional generation, achieving remarkable results (Zhang et al., 2024; Yang et al., 2024; Li et al., 2025). In contrast to these works, Neural Radiance Fields (NeRFs) (Mildenhall et al., 2021) jointly model geometry and texture using implicit representations, enabling rendering from arbitrary views via volume ray casting. NeRFs are especially popular for 3D generative methods that only rely on image data for supervision, such as DreamFusion (Poole et al., 2022).

**Signed Distance Functions (SDF)**   Signed Distance Functions (SDFs) provide a scalar field where each point in space is assigned a value representing its signed distance to the closest surface, with negative values indicating points inside the object. SDFs offer a more expressive alternative to occupancy functions, capturing both the geometry and topology of 3D shapes. DeepSDF (Park et al., 2019) pioneered the use of neural networks to learn continuous SDF representations from data. Building on this, DISN (Xu et al., 2019) introduced an amortized approach that eliminates the need for test-time optimization, enabling more efficient inference. SDFs can be discretized over a voxel grid, resulting in a *sampled SDF* or *grid SDF*, which facilitates the use of convolutional architectures. SDFusion (Cheng et al., 2023) demonstrated the effectiveness of this approach for 3D generation. Additionally, Mittal et al. (2022) proposed AutoSDF, which generates SDF grids for shape patches autoregressively after encoding them with a VQ-VAE. Further advancements include methods like 3D-LDM (Nam et al., 2022), where an MLP autoencoder predicts SDF values from latent codes and query points, blending occupancy and SDF representations. Xie et al. (2024) compared different representations, concluding that tensorial SDFs outperform triplane SDFs and tensorial NeRFs in terms of fidelity and efficiency.

**Triplanes**   Triplane representations encode 3D scenes using three orthogonal feature planes $(\boldsymbol{xy}, \boldsymbol{xz}, \boldsymbol{yz}) \in \mathbb{R}^{N \times N \times C}$, significantly reducing memory requirements while retaining spatial information. This approach was popularized by EG3D (Chan et al., 2022), which employs a triplane representation within a Generative Adversarial Network (GAN) framework for high-quality 3D-aware image synthesis. Recent works have extended triplane representations to 3D generative modeling. Approaches such as 3DGen Gupta et al. (2023), RODIN Wang et al. (2023), Blockfusion Wu et al. (2024b) and Direct3D Wu et al. (2024a) rely on latent diffusion to denoise and then up-sample triplane latents, which are then decoded via a lightweight MLP occupancy network and rendered volumetrically, facilitating the generation of high-fidelity 3D avatars or even complete 3D environments. Practically, triplanes can be obtained from occupancy data by pretraining a dedicated PointNet-UNet-OccNet autoencoder on a reconstruction loss, as detailed in Peng et al. (2020). In this approach, the triplane latent resolution can be adjusted using specific UNet layers within the autoencoder. Alternatively, a triplane dataset can be directly optimized from a mesh dataset, as proposed in Wu et al. (2024b), by jointly training an MLP decoder and its triplane input—considered here as an optimization variable on par with the weights and biases of the MLP decoder—on a reconstruction loss over the entire dataset. In this context, since the triplane equivalent of each asset in the database is readily available after pretraining, the encoder part of the generative approach consists only of projection layers. These layers encode high-dimensional triplanes into lower-dimensional triplane latents, on which the denoising diffusion model will subsequently act.

Apart from reconstruction and generation tasks, 3D representations were leveraged for CLIP-like foundation models that learn an encoding of 3D objects, which is used for downstream tasks such as classification or segmentation Zhou et al. (2023); Hegde et al. (2023).

## B   Generative pipelines

Diffusion models (Sohl-Dickstein et al., 2015) have emerged as a powerful class of generative models, demonstrating remarkable success in image synthesis (Ho et al., 2020). They operate by progressively corrupting training data through the sequential addition of Gaussian noise (forward process) and then learning to recover the original data by reversing this noising process (reverse process). This framework has been extended to 3D data, allowing for the generation of complex 3D structures (Lyu et al., 2021; Zhou et al., 2021).

**Forward Process**   Given a data sample $\boldsymbol{x}_0$ drawn from a distribution $q\left(\boldsymbol{x}_0\right)$, the forward diffusion process generates a sequence of increasingly noisy samples $\boldsymbol{x}_t$, $\forall t \in [1, \cdots, T]$ from $\boldsymbol{x}_0$ by adding Gaussian noise $\boldsymbol{\varepsilon} \sim \mathcal{N}\left(\mathbf{0}, \boldsymbol{I}\right)$ at each timestep according to a predefined variance schedule $0 < \beta_1 < \cdots < \beta_T < 1$:

$$q(\boldsymbol{x}_{1:T}|\boldsymbol{x}_0) \quad := \quad \prod_{t=1}^{T} q\left(\boldsymbol{x}_t|\boldsymbol{x}_{t-1}\right), \tag{1}$$

$$q\left(\boldsymbol{x}_t|\boldsymbol{x}_{t-1}\right) \quad := \quad \mathcal{N}\left(\sqrt{1-\beta_t}\boldsymbol{x}_{t-1}, \beta_t\boldsymbol{I}\right). \tag{2}$$

A relevant property of such a diffusion process is that $\boldsymbol{x}_t$ can be sampled from $\boldsymbol{x}_0$ using the closed-form expression:

$$q(\boldsymbol{x}_t|\boldsymbol{x}_0) = \mathcal{N}(\sqrt{\bar{\alpha}_t}\boldsymbol{x}_0, (1-\bar{\alpha}_t)\,\boldsymbol{I}), \tag{3}$$

where $\alpha_t := 1 - \beta_t$ and $\bar{\alpha}_t = \prod_{s=1}^{t} \alpha_s$. This leads to the practical sampling equation:

$$\boldsymbol{x}_t = \sqrt{\bar{\alpha}_t}\boldsymbol{x}_0 + \sqrt{1-\bar{\alpha}_t}\boldsymbol{\varepsilon}, \quad \boldsymbol{\varepsilon} \sim \mathcal{N}(\mathbf{0}, \boldsymbol{I}). \tag{4}$$

**Reverse Process.**   Initiating from a standard Gaussian distribution, $\boldsymbol{x}_T \sim \mathcal{N}(0, \boldsymbol{I})$, a denoising model $p_\theta$ parameterized by trainable weights $\theta$, learns to approximate a series of Gaussian transitions $p_\theta\left(\boldsymbol{x}_{t-1}|\boldsymbol{x}_t\right)$. These transitions incrementally denoise the signal such that

$$p_\theta\left(\boldsymbol{x}_{0:T}\right) \quad := \quad p_\theta\left(\boldsymbol{x}_T\right)\prod_{t=1}^{T} p_\theta\left(\boldsymbol{x}_{t-1}|\boldsymbol{x}_t\right), \tag{5}$$

$$p_\theta\left(\boldsymbol{x}_{t-1}|\boldsymbol{x}_t\right) \quad := \quad \mathcal{N}\left(\boldsymbol{\mu}_\theta\left(\boldsymbol{x}_t, t\right), \boldsymbol{\Sigma}_\theta\left(\boldsymbol{x}_t, t\right)\right). \tag{6}$$

Following the approach from Ho et al. (2020), we define

$$\boldsymbol{\mu}_\theta \quad = \quad \frac{1}{\sqrt{\alpha_t}}\left(\boldsymbol{x}_t - \frac{\beta_t}{\sqrt{1-\bar{\alpha}_t}}\boldsymbol{\varepsilon}_\theta\left(\boldsymbol{x}_t, t\right)\right) \tag{7}$$

$$\boldsymbol{\Sigma}_\theta(\boldsymbol{x}_t, t) \quad = \quad \sigma_t^2\boldsymbol{I} \tag{8}$$

yielding the following Langevin dynamics

$$\boldsymbol{x}_{t-1} = \frac{1}{\sqrt{\alpha_t}}\left(\boldsymbol{x}_t - \frac{\beta_t}{\sqrt{1-\bar{\alpha}_t}}\boldsymbol{\varepsilon}_\theta\left(\boldsymbol{x}_t, t\right)\right) + \sigma_t\boldsymbol{z}, \tag{9}$$

where $\boldsymbol{z} \sim \mathcal{N}\left(\mathbf{0}, \boldsymbol{I}\right)$ and $\boldsymbol{\varepsilon}_\theta\left(\boldsymbol{x}_t, t\right)$ is a learnable network approximating the per-step noise on $\boldsymbol{x}_t$.

**Loss Function**   The model is trained by minimizing the variational bound on negative log-likelihood. However, Ho et al. (2020) showed that a simplified loss focusing on the noise prediction yields good empirical results:

$$\mathcal{L} = \mathbb{E}_{\boldsymbol{x}_0, \boldsymbol{\varepsilon}_t}\left[\left\|\boldsymbol{\varepsilon}_t - \boldsymbol{\varepsilon}_\theta\left(\sqrt{\alpha_t}\boldsymbol{x}_0 + \sqrt{1-\bar{\alpha}_t}\boldsymbol{\varepsilon}_t, t\right)\right\|^2\right] \tag{10}$$

This loss encourages the network to predict the noise added at each timestep, facilitating the denoising process during generation.

We adopt open-source implementations for DiT[3] and a 3D U-Net[4]. Specific parameter settings will be provided as `hydra` config files with our open-source codebase.

## C  Evaluation Metrics

We use the following metrics for evaluating our approach. The symmetric Chamfer distance is selected to measure the distance between two point clouds $X$ and $Y$.

$$\text{CD}(X,Y) = \frac{1}{|X|} \sum_{\mathbf{x} \in X} \min_{\mathbf{y} \in Y} \|\mathbf{x} - \mathbf{y}\|_2 + \frac{1}{|Y|} \sum_{\mathbf{y} \in Y} \min_{\mathbf{x} \in X} \|\mathbf{y} - \mathbf{x}\|_2. \tag{11}$$

The F-score is the harmonic mean of precision and recall for a generated mesh $\mathcal{G}$ and a reference mesh $\mathcal{R}$. The precision is defined as

$$P(\tau) = \frac{100}{|\mathcal{G}|} \sum_{\mathbf{g} \in \mathcal{G}} \left[ \min_{\mathbf{r} \in \mathcal{R}} \|\mathbf{g} - \mathbf{r}\|_2 < \tau \right], \tag{12}$$

with $\mathbf{g}$ as points sampled on the surface of the generated mesh, $\mathbf{r}$ as points sampled from the reference mesh, $\tau$ as a threshold, and $[\cdot]$ as the Iversion bracket. The recall is defined accordingly as

$$R(\tau) = \frac{100}{|\mathcal{R}|} \sum_{\mathbf{r} \in \mathcal{R}} \left[ \min_{\mathbf{g} \in \mathcal{G}} \|\mathbf{r} - \mathbf{g}\|_2 < \tau \right]. \tag{13}$$

The final F-score is then computed as

$$F(\tau) = \frac{2P(\tau)R(\tau)}{P(\tau) + R(\tau)}. \tag{14}$$

We define the normal consistency between a generated mesh $\mathcal{G}$ and a reference mesh $\mathcal{R}$ as

$$\text{NC}(\mathcal{G}, \mathcal{R}) = \frac{1}{|\mathcal{G}|} \sum_{\mathbf{g} \in \mathcal{G}} \langle \mathbf{n}(\mathbf{g}), \mathbf{n}(N_{\mathbf{r}}) \rangle. \tag{15}$$

The function $\mathbf{n}(\cdot)$ retrieves the normal of the respective point, $\mathbf{g}$ is a point sampled on the surface of the generated mesh and $N_{\mathbf{r}}$ is the closest point on $\mathcal{R}$ to the point $\mathbf{g}$. $\langle \cdot, \cdot \rangle$ is the dot product which measures the similarity of the normals.

To measure the quality of the generated meshes w.r.t. a set of reference meshes we use the metrics Coverage (COV), Minimum matching distance (MMD), and 1-nearest neighbor accuracy (1-NNA).

$$\text{COV}(S_g, S_r) = \frac{|\{\arg\min_{Y \in S_r} D(X, Y) | X \in S_g\}|}{|S_r|}. \tag{16}$$

$$\text{MMD}(S_g, S_r) = \frac{1}{|S_r|} \sum_{Y \in S_r} \min_{X \in S_g} D(X, Y). \tag{17}$$

$$\text{1-NNA}(S_g, S_r) = \frac{\sum_{X \in S_g} \mathbb{I}[N_x \in S_g] + \sum_{Y \in S_r} \mathbb{I}[N_y \in S_r]}{|S_g| + |S_r|}. \tag{18}$$

$N_X$ is the nearest neighbor of $X$ in the set $S_r \cup S_g - \{X\}$. $S_r$, $S_g$ are the sets of reference and generated meshes and are of equal size. We use the Chamfer distance for $D$ in all our experiments.

---

[3]https://github.com/facebookresearch/DiT
[4]https://github.com/CompVis/latent-diffusion/

## D  Training Details

All training parameters can be found in the config files in the `configs` directory in the root of the repository. The main entry point is the `configs/train.yaml` files. For specific models and experiments, training parameters can be found in the respective `configs/model` and `configs/experiment` subfolders. For instance, the configuration for the generation experiment with the SDF representation using an autoencoder and diffusion transformer on the Airplane category is in `configs/experiment/diffuse_sdf_ae_dit_airplane.yaml`.

We give a summary of the most important training parameters here. We trained all models using Adam as optimizer until convergence. The learning rate and schedule highly depends on the model used and have been found by hyperparameter search. For most experiments, we use a step-wise exponential decay learning rate schedule. For Shape2VecSet and Triplane models, we use a half-cycle cosine decay started after a warm-up phase. We have trained the models using up to 2 NVIDIA A6000 GPUs. Training times for all models are within 5 days.

## E  Reconstruction and generation results on Objaverse data

To validate our results on a second widely-used dataset, we repeat the analysis on Objaverse data. We selected five categories for which most category-labeled samples were available: chair, seashell, apple, banana, mug, shield and skateboard (together 876/113/183 train/val/test samples). We first train the encoders, then the diffusion models on the same data with the same split.

The reconstruction results are shown in Table 4. These results mostly align with the ShapeNet results: 1) The value ranges are similar. 2) They confirm SDF AE as the best method for reconstruction. 3) The ranking of methods mostly aligns (compare Table 3). The only change for Objaverse is a better performance for DualOctree, surpassing Shape2vecset; however, these methods had similar F-score for ShapeNet (F-score of 79 for Shape2VecSet vs 76 for DualOctree).

Table 4: Reconstruction performance of best generation approaches on the Objaverse dataset.

| Representation | Encoder | F-score (0.0125) ($\tau = \frac{1}{80}$) | CD ($*1e-4$) | NC |
|---|---|---|---|---|
| DualOctree | VAE | $83.86 \pm 15.173$ | $0.016 \pm 10.51$ | $0.857 \pm 0.081$ |
| SDF | AE | $96.423 \pm 6.107$ | $0.009 \pm 4.948$ | $0.896 \pm 0.062$ |
| Shape2VecSet | AE | $67.578 \pm 28.472$ | $0.045 \pm 55.053$ | $0.815 \pm 0.128$ |
| Triplane | AE | $66.883 \pm 21.589$ | $0.036 \pm 41.114$ | $0.82 \pm 0.092$ |
| Voxel | AE | $96.878 \pm 6.935$ | $0.008 \pm 7.339$ | $0.85 \pm 0.06$ |

Furthermore, Table 5 shows the metrics for unconditional generation. Only the Triplane results improved considerably compared to the experiments on ShapeNet, surpassing Shape2Vecset in the Objaverse experiments. The main result, putting DualOctree and SDF as the best methods, remains the same.

It is worth noting that we computed the metrics using 183 samples for the generated and reference set respectively, since this is the size of the test set. As shown in Figure 2, low sample size biases the results. Indeed, we see higher coverage and lower 1-NNA than for our evaluation on ShapeNet, where we evaluated on 400 samples.

Table 5: Evaluation metrics for unconditional generation trained on Objaverse dataset including COV, MMD, and 1-NNA scores.

| Representation | Generator | Encoder | COV | MMD | 1-NNA |
|---|---|---|---|---|---|
| DualOctree | VAE | UNet | 0.47541 | 0.0237217 | 0.584699 |
| SDF | AE | DiT | 0.491803 | 0.0277514 | 0.68306 |
| Shape2VecSet | AE | DiT | 0.349727 | 0.0298354 | 0.819672 |
| Triplane | AE | UNet | 0.47541 | 0.0248811 | 0.729508 |
| Voxel | AE | DiT | 0.409836 | 0.0371019 | 0.803279 |

## F  Ablation study: Reconstruction performance

Table 6 shows the results for reconstruction for the ShapeNet Chair category across all tested encoders. In most cases, a standard autoencoder (with LayerNorm) performs best. This is expected since penalizing the KL-divergence in VAEs stirs the distribution at the cost of lower reconstruction performance. However, we find that the encoder choice only plays a minor role compared to the differences between representations. Specifically, the inter-representation standard deviation (between best CD per representations) is 0.0098 whereas the intra-representation standard deviation (between encoder-wise CD for each representation) is only 0.0050[5].

| Representation | Encoder | F-score ($\tau = \frac{1}{80}$) | CD (*1e-4) | NC |
|---|---|---|---|---|
| DualOctree | AE | 91.629 ± 7.5 | 0.012 ± 0.01 | 0.821 ± 0.07 |
| DualOctree | VAE | 83.152 ± 12.38 | 0.017 ± 0.01 | 0.798 ± 0.07 |
| DualOctree | VQVAE | 73.763 ± 14.23 | 0.023 ± 0.01 | 0.773 ± 0.07 |
| NeRF | AE | 58.162 ± 13.74 | 0.032 ± 0.02 | 0.714 ± 0.07 |
| NeRF | VAE | 57.326 ± 15.69 | 0.033 ± 0.02 | 0.786 ± 0.07 |
| SDF | AE | 94.332 ± 6.69 | 0.011 ± 0.01 | 0.847 ± 0.06 |
| SDF | VAE | 92.497 ± 7.05 | 0.013 ± 0.0 | 0.842 ± 0.06 |
| SDF | VQVAE | 94.355 ± 6.37 | 0.011 ± 0.01 | 0.84 ± 0.06 |
| Shape2VecSet | AE | 85.127 ± 13.81 | 0.019 ± 0.01 | 0.812 ± 0.06 |
| Triplane | AE | 63.666 ± 17.73 | 0.033 ± 0.03 | 0.761 ± 0.08 |
| Triplane | VAE | 43.952 ± 14.02 | 0.052 ± 0.04 | 0.733 ± 0.08 |
| Triplane | VQVAE | 45.216 ± 14.22 | 0.046 ± 0.03 | 0.733 ± 0.08 |
| Voxel | AE | 90.433 ± 13.14 | 0.016 ± 0.02 | 0.821 ± 0.07 |
| Voxel | VAE | 90.5 ± 12.93 | 0.016 ± 0.02 | 0.821 ± 0.07 |
| Voxel | VQVAE | 90.408 ± 13.03 | 0.016 ± 0.02 | 0.821 ± 0.07 |

Table 6: Reconstruction performance by encoder (solely ShapeNet-Chair category).

Furthermore, Figure 8 shows qualitative results for reconstruction. Thin or delicate structures lead to visible errors as missing parts of the object or as loss of details. This error mode is common for all methods and includes grid-less methods like NeRF.

## G  Ablation study: Generation performance

Table 7 compares all variants of the generative model. The applicability of DiT and Unet depends significantly on the structure of the latent vector. In most cases, AE outperforms VQ and VQVAE. DiT is more suitable for NeRF, Voxel grid and Shape2VecSet, whereas U-Net yields better results with DualOctree, SDF grid and Triplane. Interestingly, DiT also works well for grid-based representations that were usually just used with U-Net based diffusion in related work. Due to the shorter training times and higher memory efficiency of DiT, this is a promising result for future model development.

Figure 9 and Figure 10 provide further qualitative results for the best generative approaches per representation.

## H  Mesh reconstruction under varying preprocessing methods

### H.1  Preprocessing non-watertight meshes

The analysis provided in subsection 4.4 shows that some mesh preprocessing methods, such as *Mesh2SDF*, substantially alter the meshes by making them thicker and thereby introduce a bias. In accordance with the

---

[5]The intra-representation StD was only computed for the representations where results are available for all three encoder models

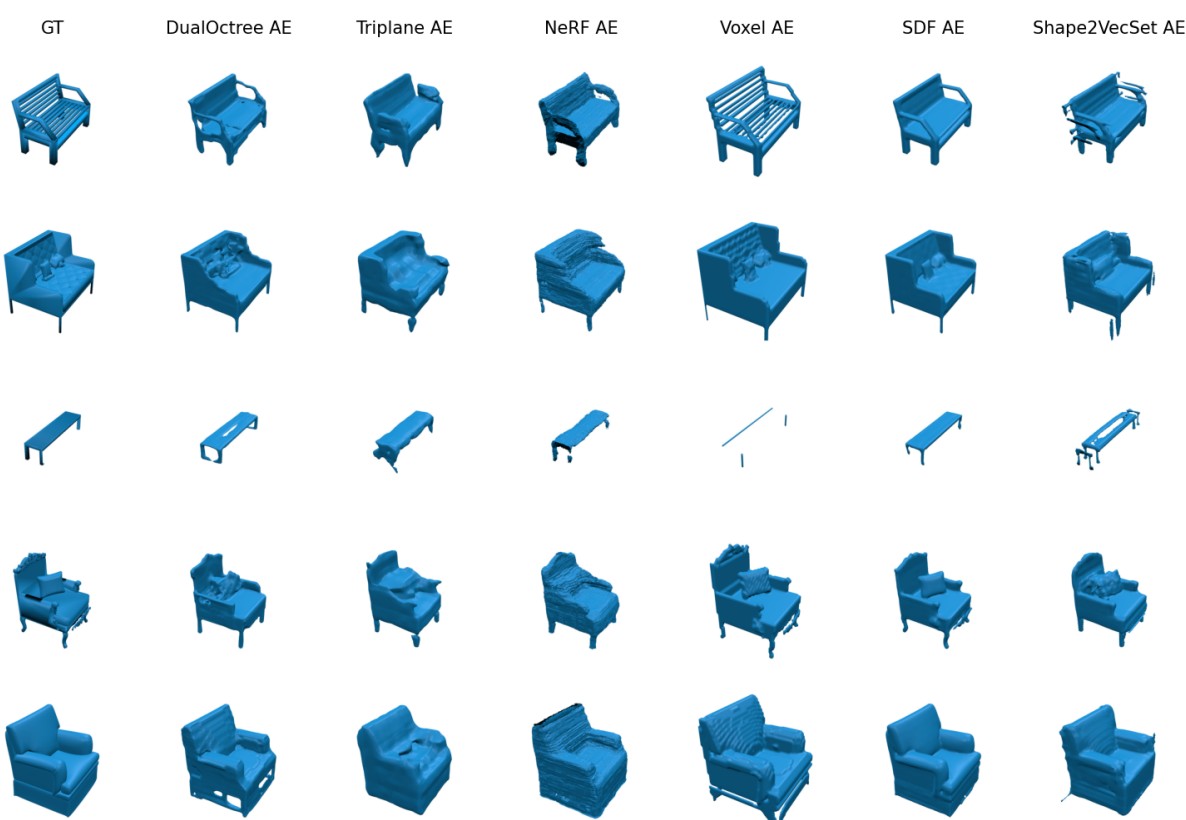

Figure 8: Qualitative results for mesh reconstruction.

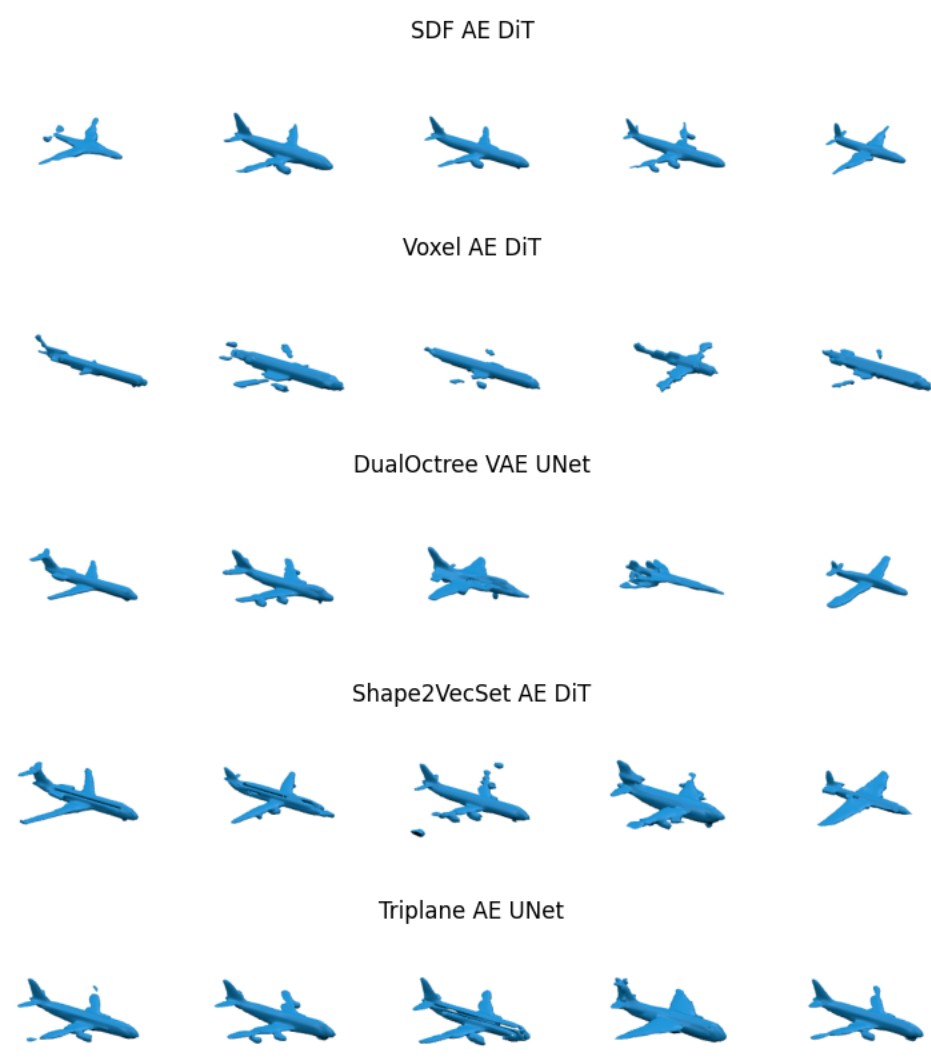

Figure 9: Qualitative results for the Airplane category

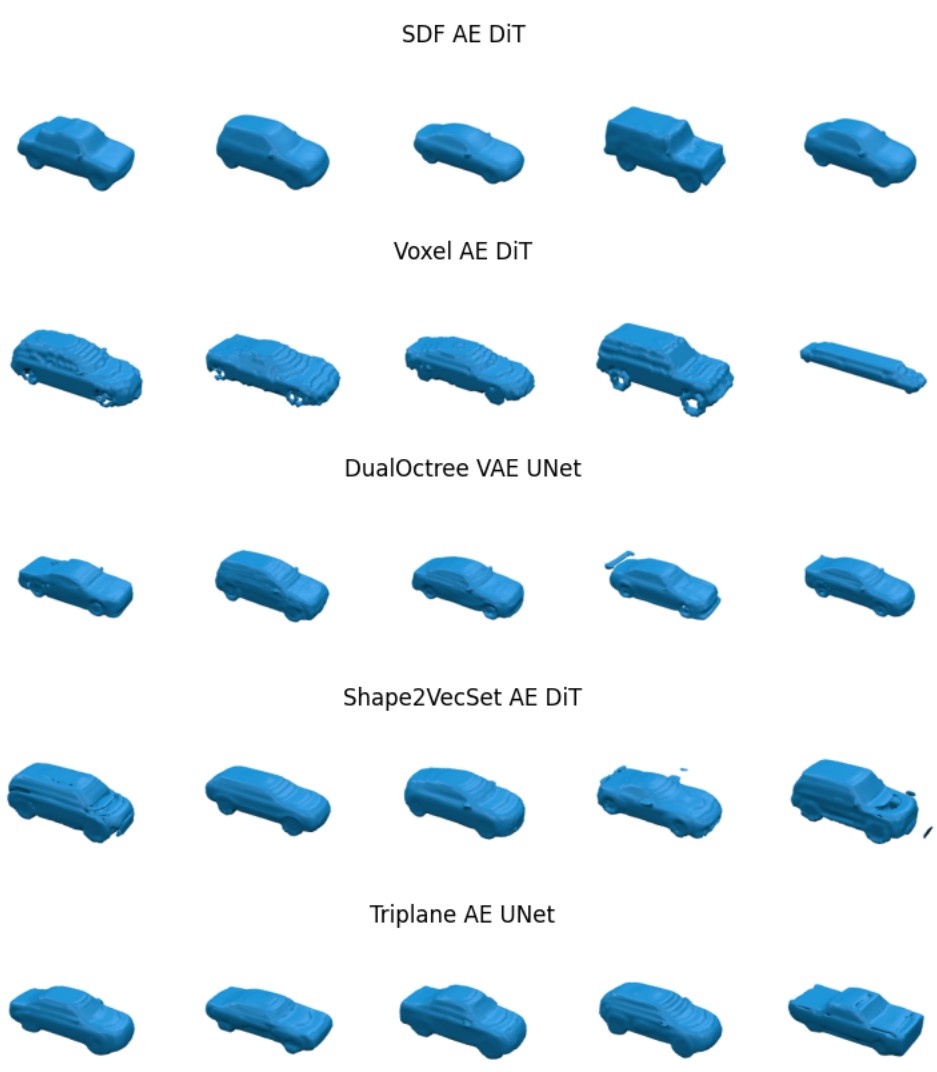

Figure 10: Qualitative results for the Car category

| Representation | Generator | Encoder | COV | MMD | 1-NNA |
|---|---|---|---|---|---|
| DualOctree | Unet | AE | 0.367 | 0.059 | 0.900 |
| DualOctree | Unet | UNet | 0.458 | 0.046 | 0.686 |
| DualOctree | Unet | VAE | 0.440 | 0.047 | 0.662 |
| DualOctree | Unet | VQVAE | 0.393 | 0.049 | 0.708 |
| NeRF | DiT | AE | 0.240 | 0.085 | 0.965 |
| SDF | DiT | AE | 0.432 | 0.048 | 0.726 |
| SDF | DiT | VAE | 0.445 | 0.050 | 0.718 |
| SDF | DiT | VQVAE | 0.385 | 0.053 | 0.728 |
| SDF | Unet | AE | 0.372 | 0.054 | 0.787 |
| SDF | Unet | VAE | 0.378 | 0.058 | 0.743 |
| SDF | Unet | VQVAE | 0.347 | 0.054 | 0.744 |
| Shape2VecSet | DiT | AE | 0.400 | 0.048 | 0.790 |
| Triplane | Unet | AE | 0.422 | 0.052 | 0.815 |
| Voxel | DiT | AE | 0.427 | 0.056 | 0.894 |
| Voxel | DiT | VAE | 0.388 | 0.066 | 0.878 |
| Voxel | DiT | VQVAE | 0.357 | 0.060 | 0.891 |
| Voxel | Unet | AE | 0.385 | 0.059 | 0.881 |
| Voxel | Unet | VAE | 0.380 | 0.058 | 0.828 |
| Voxel | Unet | VQVAE | 0.438 | 0.061 | 0.826 |

Table 7: Ablation study on generation performance (only "Chair" category). Combinations that did not converge to a state of proper 3D model generation are left out.

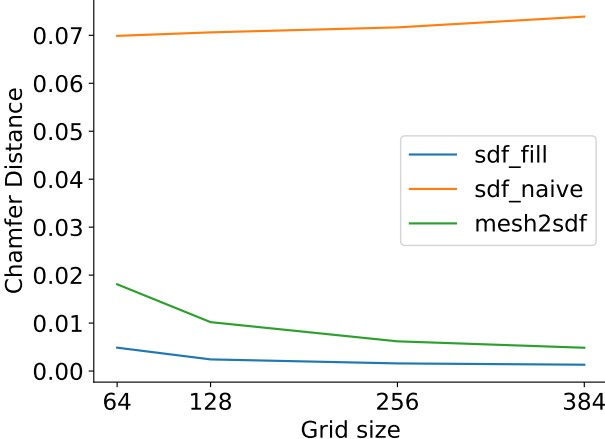

Figure 11: Comparison of different mesh conversion methods in terms of Chamfer distance.

F-score comparison in Figure 7b, the comparison of CD provided in Figure 11 shows that our preprocessing step yields a mesh that diverges less from the original data.

Figure 12 illustrates the reconstruction of an airplane from an SDF grid with a resolution of $64^3$. We first apply the flood-fill algorithm to ensure watertightness. Then, we either directly transform the mesh into a sampled SDF and back to a mesh (Figure 12b) or apply *Manifold* and then transform and reconstruct the mesh (Figure 12c).

Without applying *Manifold*, the limited grid resolution fails to capture thin structures such as airplane wings accurately, leading to incomplete or distorted reconstructions. Conversely, manifoldizing the mesh ensures that thin structures are represented as solid volumes, allowing grid-based methods to capture these features within the constraints of the grid resolution. However, this comes at the expense of altering the original mesh geometry, which may not be desirable in applications requiring high fidelity.

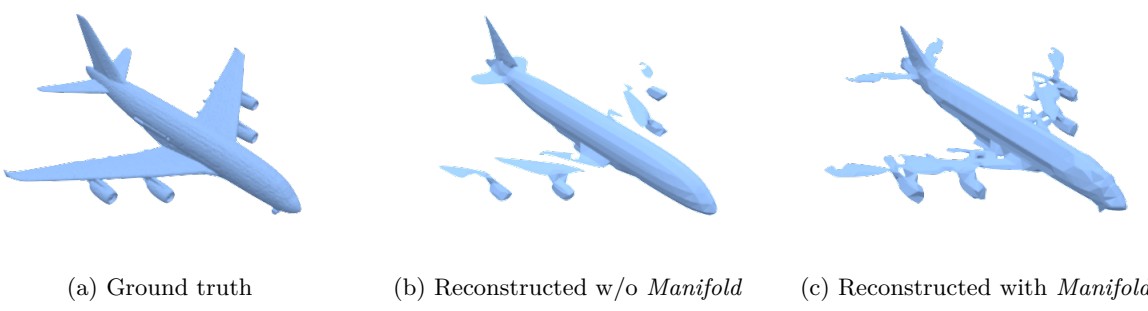

(a) Ground truth          (b) Reconstructed w/o *Manifold*          (c) Reconstructed with *Manifold*

Figure 12: Reconstruction quality when converting to a sampled SDF and back

## H.2   Dataset quality comparison

This reconstruction analysis can not only be used to compare preprocessing *methods*, but also to compare the quality of *datasets*. We analyze ShapeNet and Objaverse with our pipeline by converting all meshes to SDF grids and then convert this representation back to the mesh format. Figure 13 shows the distribution of reconstruction errors measured here as F-score. We observe that a larger fraction of objects in Objaverse is of lower quality, i.e., a conversion to an implicit representation produces a larger error.

Further, we can observe on the ShapeNet dataset that quality differs significantly between categories. Figure 14 shows the reconstruction error for the three categories *airplane, car, chair*. The F-scores for the *car* category are low even for a grid of size $384^3$, which we attribute to internal structures not captured by our preprocessing method.

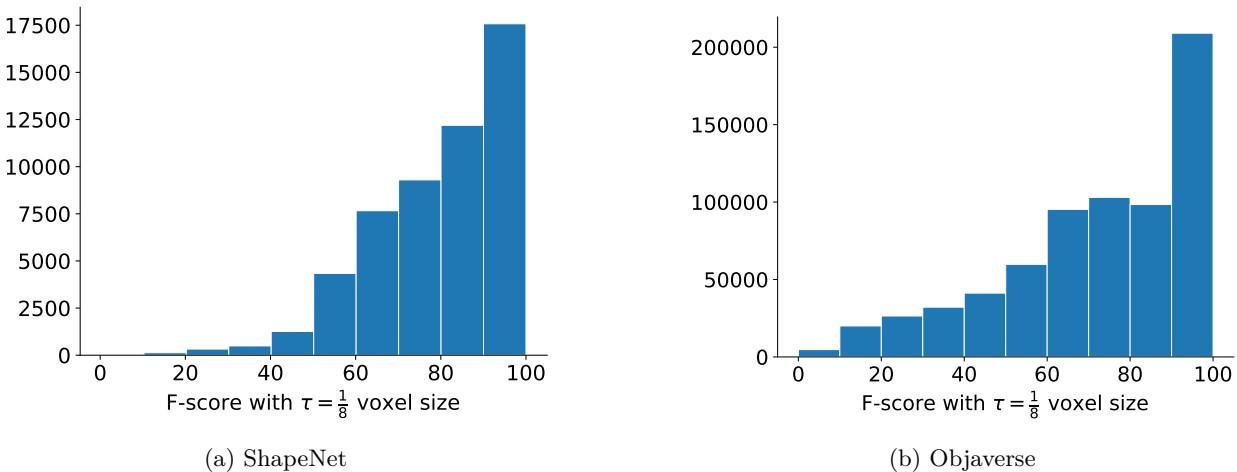

(a) ShapeNet                                    (b) Objaverse

Figure 13: Distribution of F-scores between original and reconstructed mesh using an SDF grid of size $384^3$. The threshold for the F-score is $\frac{1}{8}$ of the size of a voxel.

## I   Representation Scalability in Complex Objects

We analyze how well representations handle complex shapes. We approached this in three ways: (1) We measure object complexity in the ShapeNet test set and relate it to the reconstruction performance, (2) We use shapes with sharp edges (cube), smooth surfaces (sphere), and complex fractal surfaces (Mandelbulb), and trained the methods to encode and reconstruct them, and (3) we evaluate the complexity of the *generated* objects to study difficulties in generating complex objects. "Complexity" in (1) and (3) is measured using triangle count and surface-to-volume ratio ($\frac{S}{V}$).

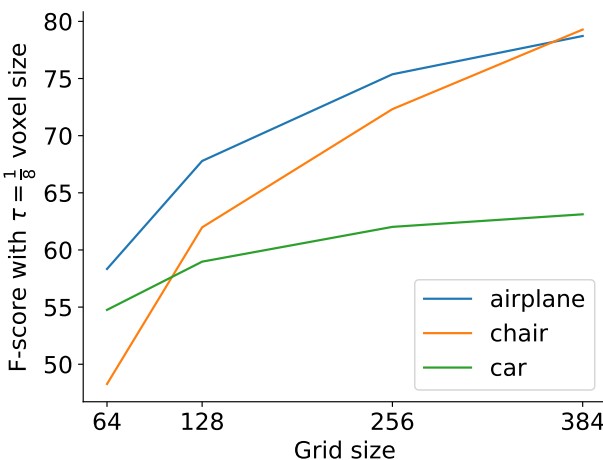

Figure 14: Effect of the grid resolution on the round trip conversion errors from mesh to SDF grid and back with our flood-fill method. Note that the threshold $\tau$ for computing the F-score is relative to the voxel size of the used grid.

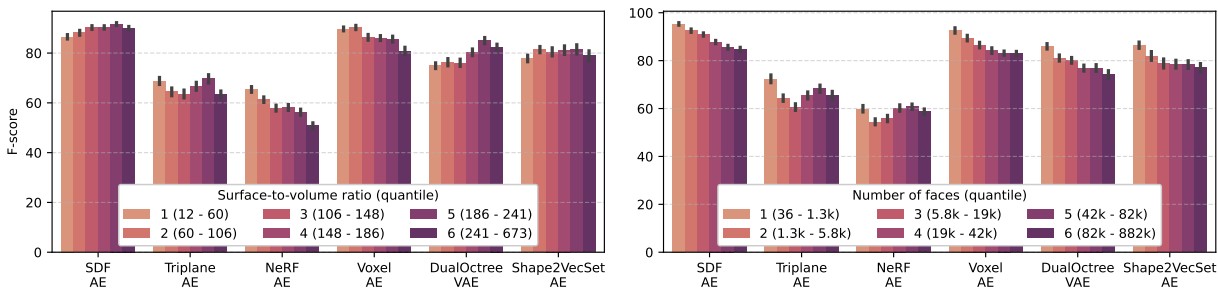

Figure 15: Complexity analysis. Best viewed zoomed in.

For approach (1), i.e. measuring object complexity in ShapeNet, Figure 15 (a) shows that voxel grids and NeRF suffer from higher $\frac{S}{V}$, whereas other representations are not affected. Figure 15 (b) shows that a higher triangle count impedes the reconstruction performance for all representations. Shape2VecSet, however, is expected to better deal with high $\frac{S}{V}$ when more points are sampled. Furthermore, our analysis of specific complex shapes (approach (2)) shows that there are failure cases when reconstructing fractal shapes. SDF-based generation smoothes the surfaces, whereas 3DShape2VecSet introduces small artifacts (see Figure 16).

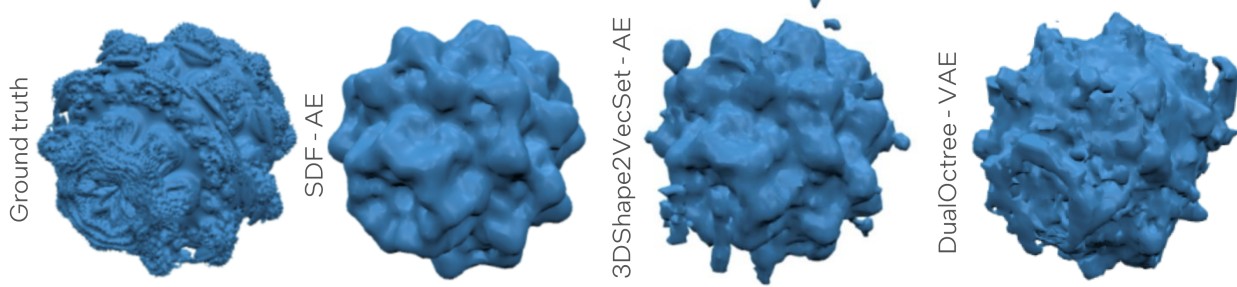

Figure 16: Mandelbulb reconstruction

Approach (3), i.e. evaluating the complexity of generated objects, finds that the $\frac{S}{V}$ is generally lower for generated objects (see Figure 17), specifically $40 \pm 18$ for SDF-generated, $61 \pm 27$ for DualOctree-generated,

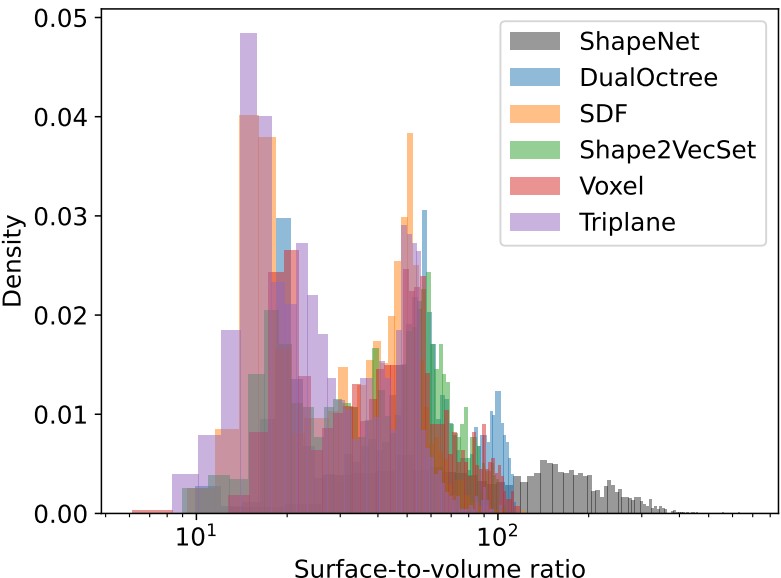

Figure 17: Surface-to-volume ratio for generated objects and ShapeNet, using categories *airplane*, *car* and *chair*.

vs $156 \pm 90$ for ShapeNet objects ($N = 400$). This suggests a bias towards generating simple, smooth objects. Training on more complex assets, e.g. Objaverse, could alleviate this issue.

## J  User Study

To capture human preferences of objects generated by different methods, we asked 24 users to rate objects generated for the chair category. Each user was shown pairs of objects and asked to select the one they preferred considering the object complexity and surface quality. To rank the methods we use the Bradley-Terry model, which models the probability of a method $A$ being better than a method $B$ as

$$P(A > B) = \frac{\exp(p_A)}{\exp(p_A) + \exp(p_B)}. \tag{19}$$

The parameters $p_A$ and $p_B$ are scores for the respective methods. We estimate the scores $p$ by minimizing the cross entropy $CE$

$$\min_{\mathbf{p}} \sum_i CE\left(\mathbf{p}, y_i\right)) \tag{20}$$

with $\mathbf{p}$ as the vector of scores and $y_i$ as the preference labels collected from the participants. The results are visualized in Figure 3 showing that objects generated with the SDF representations are preferred over the other representations.

