# OpenReview forum: "Unifi3D: A Study on 3D Representations for Generation and Reconstruction in a Common Framework"
_TMLR — Accepted by TMLR_

### Review · Reviewer_RxWz · 2025-06-08

**Summary Of Contributions:**

The authors implement a latent diffusion based generation pipeline to analyze different 3D representations. The authors analyze the reconstruction as well as generation capability. In addition, the authors provide some insights for common problems and derive best practices. The authors also provide an open-source codebase for the future research.

**Audience:**

Yes

**Broader Impact Concerns:**

No obvious concerns on the ethical implications of this work.

**Claims And Evidence:**

No

**Requested Changes:**

1. For the Coverage (COV) metric in section 3, the equation (in the appendix) is not aligned with the text description. In Eqa. (16) there is no distance threshold while in the text description the authors mention "within a certain distance".
2. For the user study result in section 4.1, it is interesting that only DualOctree-based method ranks significantly different.
It seems the user study and these metrics are evaluating different aspects. If I understand it correctly, user study is comparing among generated approaches while the metrics are evaluating each approach towards ground truth. A more fair comparison (or to say, closer setting) would be also providing the ground truth to the users.
3. In Figure 4 qualitative results it is not clear why different representations are not using the same setting. For example, DualOctree is using VAE while others are using AE.
4. For Figure 5 and its corresponding analysis in section 4.2, the authors could explain a little bit more and provide some clear conclusion.
5. Section 4.4 provides some results (Figure 7) but the detailed approach is in Appendix, which would confuse the readers. The reviewer suggest detailing some approaches here, or moving the results to the Appendix with the approach.

**Strengths And Weaknesses:**

Strength:
1. The paper is well-written, with clear structure and illustration.
2. The representations covered are comprehensive and clearly introduced. The proposed pipeline is clear, and easy to adapt to new representations.

Weakness:
1. The proposed pipeline is limited to the latent diffusion setting. Although a new representation could be easily adapted, the latent-diffusion setting limits the scope of analysis.
2. The evaluation is only conducted on ShapeNet. It is possible the evaluation results are largely affected by the dataset bias. The authors should provide the similar evaluation on at least one more dataset and compare the results.
3. In the conclusion, the authors state "the errors resulting from mesh reconstruction and representation encoding alone should be reported". However, it is not clear how separated reported errors could benefit the generation task.
4. Some claims are not clearly explained or supported with evidence. For example, the authors state in the limitation section that "Consistent results across ShapeNet suggest similar outcomes would apply to Objaverse", which is not supported by experiments on Objaverse. Also, in the conclusion part, the authors suggest that "Results should not be distorted by derivatives obtained following the application of methods like Manifold, and avoid metrics that require additional preprocessing," which the reviewer do not find strong support other than Figure 10.

---

> ### Author Response · Authors · 2025-07-25
> **Response to Reviewer RxWz**
>
> Thank you for the constructive feedback. We address your points in the following:
>
> **W1 (limited to diffusion)**: We agree that our pipeline is not comprehensively covering the whole field of diverse 3D generation methods. We have commented on SDS-loss-based approaches in the paper, and we argue that GANs are not required to compare SOTA approaches as they have lost in importance. Thus, the only missing method from our point of view are autoregressive models. We have thus started implementing an autoregressive method in our code base, adapted from Octree Transformer (Ibing et al, 2023). Due to the modular setup of our code base, it will be easy to use the autoregressive model instead of DiT (once fully implemented, it is only about changing a single parameter in a config file).  We will add results on autoregressive generation to our paper.
>
> **W2 (limited to ShapeNet)**:  ShapeNet is very suitable for benchmarking as it is of higher quality than other datasets like Objaverse (see Section G.2 and Figure 11) and provides category labels, allowing for unconditional generation within a category. It is also worth noting that we train on three different categories and the results are consistent over these three experiments, providing evidence that the evaluation results are generally valid.
>
> Nevertheless, we conducted a new experiment on Objaverse data to validate our results on a second dataset. For details on the experimental setup and **for the result tables, please see our reply to Reviewer fZtW (W1)**. In summary, we find:
> * Reconstruction: As for ShapeNet, the SDF autoencoder is the strongest reconstruction method. The metric values have similar range, and the ranking of methods is the same, except for better performance of DualOctree, now outperforming the Shape2Vecset AE.
> * Generation: Again, we find very similar results for Objaverse as for ShapeNet, confirming DualOctree as the best method. The metric values are slightly different due to sample size biases (see Figure 2).
>
> We believe this experiment enriches our paper. We now also support Objaverse in our code base, thereby simplifying future benchmarking.
>
> **W3 (separately reported errors)**: Thank you for your interesting question. We believe that separately reported errors would benefit the transparency of papers and thereby the opportunity of other papers to improve specific aspects of one pipeline. For example, if everyone reported the mesh reconstruction error, e.g. the error stemming from marching cubes, future work could test other mesh reconstruction methods and easily quantify the improvement over the original pipeline.
>
> **W4 (claims)**: Regarding Objaverse, please see above. Regarding the distortion of results: The evidence for this statement can be found in Figure 6, showing a considerable influence of the mesh reconstruction method on the error, and in Figure 7, showing that there are significant differences between the mesh reconstruction performances for naïve vs sdf2mesh vs our method. We have added references to these figures in the statement in the conclusion.
>
> **RQ 1 (COV metric)**: Thank you for pointing this out. We fixed the description: “Coverage measures the fraction of $S_r$ that is matched to $S_g$”.
>
> **RQ2 (user study)**: We believe this is a misunderstanding. As we are evaluating unconditional generation in both cases, there is no ground truth. The metrics are described in section D, and they are all computed between a set of generated objects and a reference set.
>
> **RQ3 (encoder model)**: The encoder model was determined in an ablation study where we tested the generative performance with AE, VAE and VQVAE for each representation. We did this study since the papers we are building up on use various encoders (e.g. 3DShape2Vecset uses AE and VAE, SDFusion uses VQVAE, etc), and we wanted to select the best one for each representation to ensure a fair comparison. Please see Appendix E and F for these experiments. We have updated the caption to clarify the selection of encoders in this figure.
>
> **RQ4 (Figure 5)**: We will extend the description and make the conclusion more explicit. The experiment shows well the different characteristics and importance of different representations. For instance, DualOctree with the smallest latent size achieves better reconstructions than Triplane with a larger latent thanks to the spatial adaptive structure but is less accurate and slower than the Voxel representation that uses a much larger latent. Selecting the right representation can therefore significantly influence the runtime, accuracy and memory requirements. Generative methods targeting CAD applications may want a representation with minimal reconstruction error, while methods running on devices with small memory may favor a representation with a small latent.
>
> **RQ5 (Section 4.4)**: Thank you for the suggestion. We will move the details from the appendix to the main paper to provide a better understanding of our method.

---

### Review · Reviewer_bY2D · 2025-06-19

**Summary Of Contributions:**

This paper presents a systematic comparison of 3D representations for reconstruction and generation in a common framework.
The steps involved in this framework are: transforming the mesh into a suitable representation, using an encoder to compress it into a latent vector, applying a denoising model to denoise it, and finally, using a decoder to decompress and transform the representation back into a mesh. 3D representations are compared on quality, computational efficiency, and generalization performance metrics. The proposed framework also provides insights into best practices for 3D representations.

**Audience:**

Yes

**Broader Impact Concerns:**

No concerns.

**Claims And Evidence:**

Yes

**Requested Changes:**

- See weaknesses.
- Additional references can be added ("3D Gaussian Splatting for Real-Time Radiance Field Rendering" by Kerbl et al., "Uni3D: Exploring Unified 3D Representation at Scale" by Zhou et al.,  "Boost Your NeRF: A Model-Agnostic Mixture of Experts Framework for High Quality and Efficient Rendering" by Di Sario et al.).

**Strengths And Weaknesses:**

### Strengths
- This work offers a "universal" framework to compare 3D representations.
- It also highlights that generation and reconstruction capability should be considered jointly during evaluation.
- The paper is well-written and organized.

### Weaknesses
- The authors should better clarify why the proposed framework is better to benchmark 3D representations instead of reconstructing/generating with the standard original recipe of the different models.
- Gaussian splatting methods are missing in the benchmark. I think the authors should incorporate 3DGS into the framework, as it's now the state-of-the-art (SOTA).
- Some sentences need to be clarified (e.g., "To ensure that the latent vector behaves well [...]", what does it mean by "behaves well?").

---

> ### Author Response · Authors · 2025-07-25
> **Response to Reviewer bY2D**
>
> Thank you for the positive feedback. We address your concerns in the following:
>
> **W1: The authors should better clarify why the proposed framework is better to benchmark 3D representations instead of reconstructing/generating with the standard original recipe of the different models.**
>
> Thank you for the suggestion. The main goal of our framework and code base is to ensure comparability. As pointed out in the introduction section, each model uses different representations, encoders, generative models and preprocessing methods, but in the end, most of them only report results on conditional generation, leaving intermediate performance (e.g. of the encoder / preprocessing method) unclear. Thus, there isn’t one “standard original recipe” per se, but rather one recipe per method, making them incomparable. To ensure proper benchmarking of 3D representations, we propose such a standard recipe and apply it to multiple representations. We have clarified this point in the introduction:
>
> *Our standardized pipeline ensures a fair comparison between different representations with respect to reconstruction and generation, instead of relying on the individual implementations of different 3D generation approaches that leverage the representations in varying manner, evaluate them with varying metrics, and oftentimes solely report conditional generation performance of the complete pipeline.*
>
> **W2:  Gaussian splatting methods are missing in the benchmark. I think the authors should incorporate 3DGS into the framework, as it's now the state-of-the-art (SOTA).**
>
> We agree that 3DGS has become an important method; however, mainly in the context of (scene) rendering and novel view synthesis. The papers mentioned by the reviewer are good examples:
> * Kerbl et al. develop a fast method for scene rendering, unsuitable for mesh generation
> * Di Sario et al similarly propose an improved rendering method to be used with NeRFs
> * Zhou et al propose a powerful encoder that is used for classification, segmentation etc, but not for 3D generation
>
> Since the main advantage of Gaussian splatting is their good rendering capability and performance in representing color and texture, it is not in the scope of our study, which focuses on direct (not SDS-loss based) generation approaches. In this domain, GS is not regarded as a SOTA method - as reviewer RxWz pointed out, the SOTA in this 3D object generation are currently Shape2vecset-based methods (e.g. Trellis, CLAY).
>
> While a few approaches exist that use GS also for direct generation (e.g. GaussianAnything), there are still major challenges involved with using them for our purposes, e.g. 1) GS usually relies on multi-view image inputs, whereas our benchmark is on unconditional generation, 2) GS usually does not generate meshes, and thus would require further postprocessing steps to allow comparison to other methods.
>
> **W3: Some sentences need to be clarified (e.g., "To ensure that the latent vector behaves well [...]", what does it mean by "behaves well?").**
>
> Thank you for pointing this out. We have corrected the sentence to “To ensure that the latent vector is bounded […]” making it more concise. Please let us know what other sentences lack understandability.
>
> **RQ1: Missing references**
>
> Thank you for these suggestions. We have added these references within a discussion on Gaussian Splatting in the introduction:
> *Similarly, Gaussian Splatting has become popular in the 3D field [references], but is not included in our study, as it is usually trained with SDS-loss methods and focuses on realistic and efficient rendering, in contrast to our assumption of meshes as the ground truth.*

---

### Review · Reviewer_fZtW · 2025-07-12

**Summary Of Contributions:**

This paper presents a unified evaluation framework across different 3D representations (voxel grids, SDFs, point clouds etc.) to assess their performance in 3D reconstruction and generation. The authors transformed ground truth meshes into given 3D representations and trained different encoding networks to compress them into vectors. The generation network is trained on the latent space. The authors evaluated the performance of both reconstruction and generation using standard metrics and user studies.

**Audience:**

Yes

**Claims And Evidence:**

Yes

**Requested Changes:**

Address the weaknesses part.

**Strengths And Weaknesses:**

Strengths:
- Thorough benchmarking of a wide range of 3D representations, including SDFs, meshes, voxel grids, vecsets etc, which would be useful for the 3D generation community.
- Proposed a unified framework that integrates all 3D representations for fair comparison and evaluation.
- Sufficient experiments of different encoders and evaluation of generation and reconstruction performance, including geometric metrics and statistic analysis and user study.
- Utilize Dual Octree graph encoding, which achieves a good performance over other representations.

Weaknesses:
- The experiments were done on simple ShapeNet datasets, which is less used with Objaverse becoming the main training datasets. Therefore, the benchmark might not be general enough.
- Some of the conclusions are a bit surprising. For example, Shape2VecSet doesn't achieve not good results, however, most of the recent SoTA models are based on vecsets. For some representations (Voxel), using AE achieves better generation performance than VAE, which also not aligns with common experience.  There are no analysis towards these points.
- Lack of training details (iterations, training efficiency, model size etc) of the encoding and generation. Lack of visualization results.
- Missing citation: [1] State of the art on diffusion models for visual computing, [2] Diffusion models for 3D generation: A survey, [3] A survey on generative diffusion models

---

> ### Author Response · Authors · 2025-07-25
> **Response to Reviewer fZtW**
>
> Thank you for the constructive feedback. We are delighted that you find our paper useful for the 3D generation community.
> We address the weaknesses in the following:
>
> **W1: The experiments were done on simple ShapeNet datasets, which is less used with Objaverse becoming the main training datasets.**
>
> We agree that Objaverse is an important dataset and we have conducted additional experiments. Since preprocessing and training on all Objaverse objects is infeasible within the rebuttal period, we selected five categories for which we had most category-labeled samples: chair, seashell, apple, banana, mug, shield and skateboard (together 876/113/183 train/val/test samples). We first train the encoders, then the diffusion models on the same data with the same split. The results are presented in the following:
>
> **Reconstruction results (corresponding to Table 3):**
> | Method          | F-score (0.05)     | F-score (0.025)     | F-score (0.0125)    | CD                 | NC                |
> |:----------------|:-------------------|:--------------------|:--------------------|:-------------------|:------------------|
> | DualOctree VAE  | 99.087 $\pm$ 2.185 | 95.348 $\pm$ 6.775  | 83.86 $\pm$ 15.173  | 0.016 $\pm$ 10.51  | 0.857 $\pm$ 0.081 |
> | SDF AE          | 99.976 $\pm$ 0.161 | 99.359 $\pm$ 2.695  | 96.423 $\pm$ 6.107  | 0.009 $\pm$ 4.948  | 0.896 $\pm$ 0.062 |
> | Shape2VecSet AE | 92.87 $\pm$ 14.437 | 82.821 $\pm$ 23.974 | 67.578 $\pm$ 28.472 | 0.045 $\pm$ 55.053 | 0.815 $\pm$ 0.128 |
> | Triplane AE     | 95.764 $\pm$ 7.599 | 85.221 $\pm$ 15.922 | 66.883 $\pm$ 21.589 | 0.036 $\pm$ 41.114 | 0.82 $\pm$ 0.092  |
> | Voxel AE        | 99.328 $\pm$ 3.232 | 98.574 $\pm$ 5.299  | 96.878 $\pm$ 6.935  | 0.008 $\pm$ 7.339  | 0.85 $\pm$ 0.06   |
>
> These results mostly align with the results in the paper: 1) The value ranges are similar. 2) We confirm SDF AE as the best method for reconstruction. 3) The ranking of methods mostly aligns. For ShapeNet, it was SDF (88) > Voxel (86) > Shape2vecset (79) > DualOctree (76) > Triplane (66). The only change for Objaverse is a better performance for DualOctree, surpassing Shape2vecset; however, these methods were not far apart before.
>
> **Generation results (compare Table 2):**
> | Method              |   COV |   MMD |   1-NNA |
> |:--------------------|------:|------:|--------:|
> | DualOctree VAE UNet | 0.475 | 0.024 |   0.585 |
> | SDF AE DiT          | 0.492 | 0.028 |   0.683 |
> | Shape2VecSet AE DiT | 0.35  | 0.03  |   0.82  |
> | Triplane AE UNet    | 0.475 | 0.025 |   0.73  |
> | Voxel AE DiT        | 0.454 | 0.037 |   0.822 |
>
> Similar to reconstruction, we find that the model ranking in Objaverse is largely the same. In detail:
>
> MMD:
> * Objaverse: DualOctree < Triplane < SDF < shape2vecset
> * ShapeNet: DualOctree < SDF < Shape2vecset < Triplane < Voxel
>
> COV:
> * Objaverse: SDF > Dualoctree = Triplane > Shape2vecset
> * Shapenet: DualOctree > SDF > Shape2vecset > Voxel > Triplane
>
> 1-NNA:
> * Objaverse: DualOctree < SDF < Triplane < shape2vecset
> * Shapenet: DualOctree < SDF < Shape2vecset < Triplane < Voxel
>
> Only Triplane results improved considerably, now surpassing Shape2Vecset. The main result, putting DualOctree and SDF as the best methods, remains the same.
>
> It is worth noting that we computed the metrics using 183 samples for the generated and reference set respectively, since this is the size of the test set. As shown in Figure 2, low sample size biases the results. Indeed, we see higher coverage and lower 1-NNA than for our evaluation on ShapeNet (400 samples).
>
> **W2: Surprising conclusions (Shape2VecSet results compared to SOTA, VAE vs AE)**
>
> Thank you for the interesting observation. Shape2vecset was used later with larger models and complex multi-resolution schemes. It is difficult to evaluate quantitatively how scalable a representation is. We opted for similar grid sizes and the basic implementation of each representation, instead of using the largest model available.
>
> Furthermore, VAE usually performs superior to AEs due to its ability to normalize the distribution of the latent space. However, we observed that adding LayerNorm as the final layer of the encoder has a similar effect and thus improves reconstruction performance at similar generation capabilities. We provide ablation studies about the encoder model choice in Appendix E and F.
>
> **W3: Lack of training details. Lack of visualization results.**
> All implementation details will be available in structured hydra config files in our open-source repo (e.g. see here for the main train config, which imports parameters from sub-configs: https://anonymous.4open.science/r/unifi3d-39CD/configs/train.yaml). We will add a summary of the most common parameters to the supplement such as the optimizer (Adam) and the learning rate schedule. We will add more qualitative examples to the supplement. Please let us know of any other visualizations that can improve the paper.
>
>  **W4: Missing citations**
> Thank you, we will add these references.

---

### Comment · Reviewer_fZtW · 2025-08-11

It seems that the authors haven't incorporated the responses into the paper.

---

> ### Author Response · Authors · 2025-08-11
>
> Thank you for notifying us of this issue. Due to prior experience with OpenReview for conferences, we were unaware that we can modify the original PDF. Apologies for this oversight.
>
> However, we have already incorporated all changes into the paper during the revision period. We have now uploaded the updated PDF. For example, Appendix F includes the new Objaverse results (reviewer fZtW and RxWz), Appendix E contains training details (reviewer fZtW), Appendix H contains further visualization results (reviewer fZtW), and the conclusion comments on Gaussian Splatting (Reviewer bY2D).
>
> Of course, our responses below also contain the changes within our point-by-point responses.
> Thanks for your consideration.

---

### Decision · Action_Editor_NHku · 2025-08-13

**Recommendation:** Accept as is

**Audience:**

Yes

**Audience Explanation:**

Yes. This paper tackles an area that is of substantial interest to many members of TMLR's audience.

**Claims And Evidence:**

Yes

**Claims Explanation:**

The paper provides a unified evaluation of representations for 3D generation and reconstruction. After the revised version was uploaded, all three expert reviewers agree that the paper has accurate claims that are supported by convincing evidence. The AE agrees.